# Environmental palaeogenomic reconstruction of an Ice Age algal population

Youri Lammers [1✉], Peter D. Heintzman [1,2] & Inger Greve Alsos [1,2]

Palaeogenomics has greatly increased our knowledge of past evolutionary and ecological change, but has been restricted to the study of species that preserve either as or within fossils. Here we show the potential of shotgun metagenomics to reveal population genomic information for a taxon that does not preserve in the body fossil record, the algae *Nannochloropsis*. We shotgun sequenced two lake sediment samples dated to the Last Glacial Maximum and reconstructed full chloroplast and mitochondrial genomes to explore within-lake population genomic variation. This revealed two major haplogroups for each organellar genome, which could be assigned to known varieties of *N. limnetica*, although we show that at least three haplotypes were present using our minimum haplotype diversity estimation method. These approaches demonstrate the utility of lake sedimentary ancient DNA (*sed*aDNA) for population genomic analysis, thereby opening the door to environmental palaeogenomics, which will unlock the full potential of *sed*aDNA.

[1] The Arctic University Museum of Norway, UiT – The Arctic University of Norway, Tromsø, Norway. [2] These authors contributed equally: Peter D. Heintzman, Inger Greve Alsos. ✉email: youri.lammers@uit.no

Palaeogenomics, the genomic-scale application of ancient DNA, is revolutionizing our understanding of past evolutionary and ecological processes, including population dynamics, hybridization, and the effects of drivers of change[1–4]. Despite extensive application to, and innovations using, body fossils[5–7], its use on another major source of ancient DNA—the environment—has been almost entirely limited to inventorying taxa through time[8–15]. However, a nuanced understanding of ecological and evolutionary dynamics requires population genomic information.

Approaches to recovering population genomic variation from multi-taxon mixtures, including modern environmental DNA and microbiomes, are still in their infancy. To detect this variation, studies either utilize extensive intraspecific genomic reference datasets[16–18] or assemble de novo genomes[19–21]. However, the application of these metagenomic approaches to sedimentary ancient DNA (sedaDNA) mixtures is complicated by its degraded nature, with molecules often <100 base pairs (bp) long and with ends impacted by cytosine deamination[22,23], which impedes de novo assembly of contigs. Furthermore, most taxa have limited reference genomes available, which can hinder mapping-based haplotype estimations. Despite these issues, recent ancient DNA studies have begun to explore population genomic diversity in cave sediments[12], and archaeological middens[24] and latrines[25].

Lake sediments provide an alternative and ideal source of sedaDNA that originates from both the catchment and the lake itself, as well as providing a stable environment for optimal aDNA preservation[26,27]. Lake sedaDNA is used to infer the taxonomic composition of past communities[27,28], regardless of whether those taxa preserve in the body fossil record. While techniques such as DNA metabarcoding allow for the inventorying of groups of organisms[29,30], either deep shotgun sequencing or target enrichment of sedaDNA is required for palaeogenomic reconstruction, which would allow for robust species-level identification[12,14,24] and the potential exploration of population genomic variation.

Andøya, an island located off the coast of northwest Norway, was partially unglaciated during the Last Glacial Maximum (LGM; Fig. 1) and has therefore been a focus of palaeoecological studies[31], especially for its potential as a cryptic northern refugium[32–34]. Studies focussing on sediment cores have reported the presence of an Arctic community during the LGM[32,35–41], including the single-celled microalgae Nannochloropsis[41]. This microalgae has a cosmopolitan distribution[42–49], with all species known from marine environments, except Nannochloropsis limnetica, which is known from freshwater/brackish habitats and comprises five varieties[45,47,48]. In sediments, Nannochloropsis has not been reported from macrofossil and pollen/spore profiles, and may therefore only be identifiable using sedaDNA[10,41,50].

In this study, we shotgun sequenced LGM sediments from Andøya that had previously been shown to contain Nannochloropsis[41], and demonstrate that N. limnetica dominates the identifiable taxonomic profile. Through reconstruction of complete chloroplast and mitochondrial genomes, we show that two variants of N. limnetica are represented. We present a method to estimate that a minimum of three haplotypes are present, based on read-linked single-nucleotide polymorphisms (SNPs) from degraded sedaDNA. We thus showcase the potential for lake sedaDNA to both recover organellar genomes and estimate past population genomic diversity for taxa that are not preserved in the body fossil record.

## Results

### Metagenomic analysis and species-level determination of Nannochloropsis.
We shotgun sequenced two LGM samples,

dated to 17,700 (range: 20,200–16,500) and 19,500 (20,040–19,000) calibrated years before present (cal yr BP), to generate 133–224 million paired-end reads, of which we retained 53–127 million sequences after filtering (Supplementary Table 1). We first sought to identify the broad metagenomic profiles of the samples and the species-level identification of Nannochloropsis from Lake Øvre Æråsvatnet.

First, for each sample, we compared two non-overlapping one-million sequence subsets of the filtered data to the NCBI nucleotide database. The taxonomic overlap between the two one-million read subsets was 88–93% within each sample, demonstrating that our subsets are internally consistent. We then merged the two subsets from each sample, which resulted in the identification of 29,500–32,700 sequences (Table 1). The majority of the identified sequences were bacterial, with 21–26% identified as Mycobacterium, although the majority of these sequences could not be identified to a specific strain. Within the eukaryotes, Nannochloropsis constituted ~20% of the assigned sequences in both samples, with ~33% of these identified as N. limnetica (Table 1; Supplementary Fig. 1 and Supplementary Data 1).

To further investigate the metagenomic profile of the samples, we aligned all filtered sequences to a nuclear genome reference panel. We mapped 310,000–680,000 sequences to the N. limnetica genome, which translates to 9.3–20.3 thousand sequences per megabase (kseq/Mb), and a nuclear genomic coverage of 0.48–1.13×. We observed a far lower relative mapping frequency to all other Nannochloropsis nuclear genomes (up to 2.5–4.8 kseq/Mb). If we consider sequences that are only mappable to a single genome, then the relative mapping frequency falls to 7.4–17 kseq/Mb for N. limnetica and up to 0.6–1.3 kseq/Mb for all other Nannochloropsis genomes. The most abundant non-Nannochloropsis eukaryotic taxon in the sequence data was human, with 2–11 thousand sequences mapped (0.7–3.4 seq/Mb). As expected from the metagenomic analyses, the next most abundant group was Mycobacterium. The relative mapping frequency was consistent across all four strains, based on both all sequences aligned (up to 1.1–1.7 kseq/Mb) and only retaining sequences unique to a strain (up to 0.6–0.9 kseq/Mb). The relative frequencies of sequences mapping to plausible and implausible eukaryotic genomes were low and comparable for non-Nannochloropsis taxa (Fig. 1c). Based on both raw counts and those corrected for genome size, this analysis therefore indicated that N. limnetica is the best represented taxon in the panel (Fig. 1c and Supplementary Data 2).

We sought to confirm whether sequences identified from the three best represented taxonomic groups (Nannochloropsis, Mycobacterium, human) were likely to be of ancient origin or to have derived from modern contamination. The sequences aligned to the human genome did not exhibit typical patterns of ancient DNA damage, which include cytosine deamination and depurination-induced strand breaks (Fig. 2 and Supplementary Fig. 2). In contrast, we find that sequences aligned to two Nannochloropsis limnetica and the Mycobacterium avium genomes exhibit authentic ancient DNA damage (Fig. 2 and Supplementary Figs. 3 and 4), with patterns that are near identical for both taxonomic groups, consistent with their preservation in the same environment of broadly contemporaneous age.

We next aligned our sequence data against two organellar reference panels. Both analyses also recovered N. limnetica as the best represented taxon, with 23,600–37,900 and 8600–14,100 sequences aligning to the chloroplast and mitochondrial genome of this taxon, respectively (Supplementary Data 3). After mapping the filtered sequence data to the Nannochloropsis chloroplast genomes individually, the number of sequences uniquely aligned to N. limnetica fell to 7700–11,300

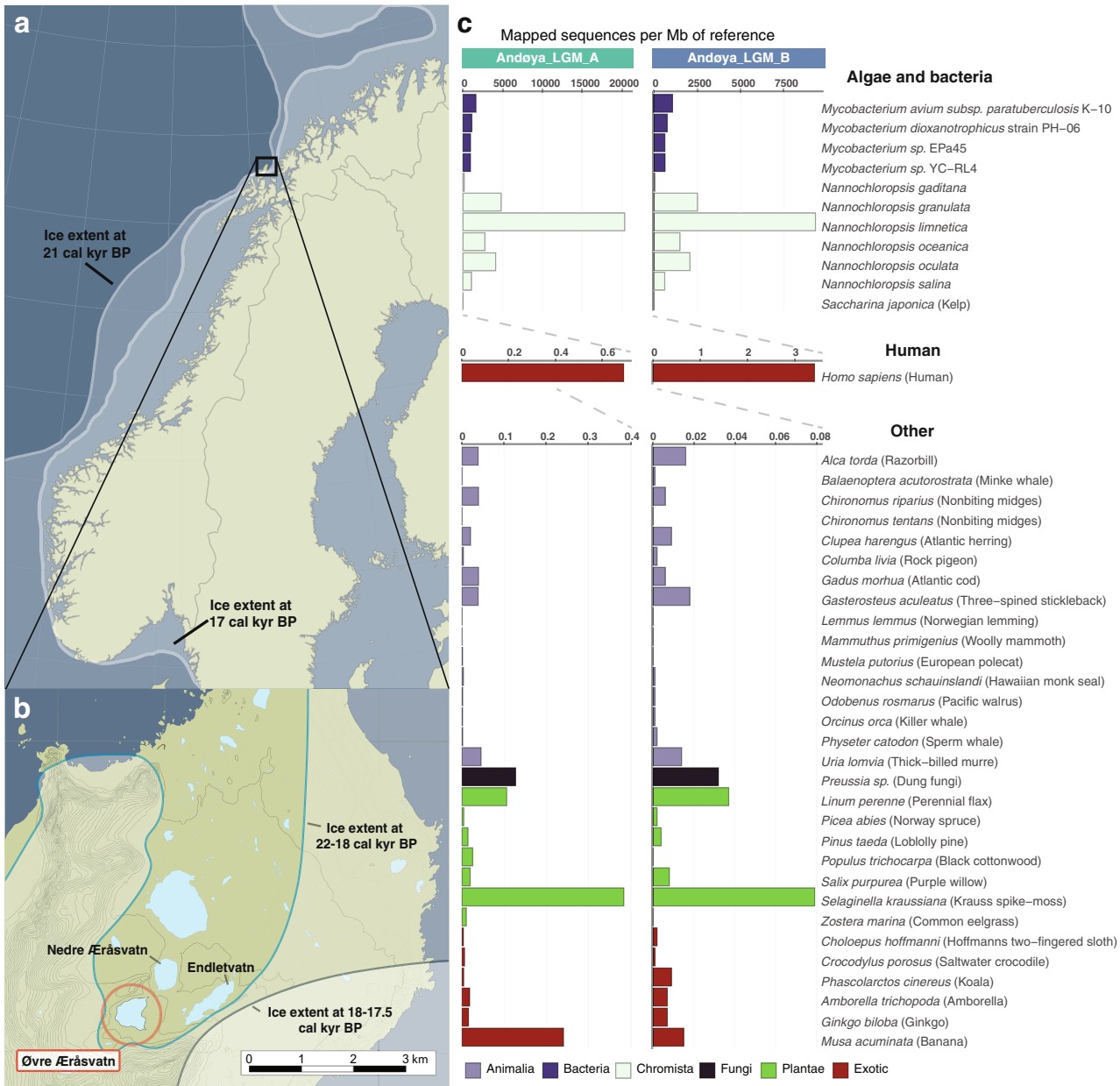

**Fig. 1 Location of Lake Øvre Æråsvatnet and taxonomic composition of the LGM Andøya sediment samples. a, b** Location of Lake Øvre Æråsvatnet (circled in red, panel **b**) in the ice-free refugium of Andøya in northwest Norway. The regional ice extent for Scandinavia in panel **a** has been plotted for 22 (outer) and 17 (inner) cal kyr BP and is based on Hughes et al.[73]. The local ice extent in panel **b** is plotted for 22–18 and 18–17.5 cal kyr BP and is based on Vorren et al.[31]. Panel **a** is made with map data from Natural Earth, while panel **b** is made with data from Kartverket.no. **c** Taxonomic composition of the LGM Andøya sediment samples, based on alignment to a reference panel of 42 eukaryotic or bacterial nuclear genomes of genera that are either expected in the region or which are exotic controls. The scale is the number of sequences mapped to each Megabase (Mb) of the reference genome. For readability, the algal, bacterial, and human results are plotted separately with differing x-axis scales.

(Supplementary Data 2). The most abundant non-*Nannochloropsis* taxon was the algae *Choricystis parasitica* with 255–1,784 and 38–276 of sequences mapped to its chloroplast and mitochondrial genome.

**Reconstruction of *Nannochloropsis* organellar palaeogenomes and their phylogenetic placement.** We reconstructed complete composite organellar palaeogenomes for *Nannochloropsis* present in LGM Andøya. The resulting complete chloroplast sequence was 117.7 kilobases (kb) in length and had a coverage depth of 64.3×. The mitochondrial genome was 38.5 kb in length with a

coverage of 62.4× (Fig. 3 and Supplementary Fig. 5 and Supplementary Table 2). We observed two major structural changes in our reconstructed chloroplast as compared to the *N. limnetica* seed sequence, in which the reconstructed chloroplast was inferred to share the ancestral structural state with the remaining *Nannochloropsis* taxa. This included a 233 bp region in a non-coding region between the *thiG* and *rpl27* genes, which is absent in the *N. limnetica* seed sequence and of varying length among all other *Nannochloropsis* taxa (Supplementary Fig. 6). A 323 bp insertion in a non-coding region between the genes *rbcS* and *psbA* is present in the *N. limnetica* seed sequence, but lacking from our reconstructed chloroplast and all other *Nannochloropsis* taxa

**Table 1 Summary of the best represented taxa (>200 identified sequences) detected in the metagenomic analysis.**

|  | Andøya_LGM_A | | | Andøya_LGM_B | | |
|---|---|---|---|---|---|---|
|  | *N* | *I* | *A* | *N* | *I* | *A* |
| Bacteria | 18,852 | 63.93 | 0.94 | 21,873 | 66.85 | 1.09 |
| *Mycobacterium* | 6268 | 21.26 | 0.31 | 8535 | 26.08 | 0.43 |
| *M. avium* complex | 444 | 1.51 | 0.02 | 635 | 1.94 | 0.03 |
| *M. dioxanotrophicus* | 0 | 0 | 0 | 229 | 0.7 | 0.01 |
| *M.* sp. EPa45 | 0 | 0 | 0 | 306 | 0.94 | 0.02 |
| *M.* sp. YC-RL4 | 0 | 0 | 0 | 207 | 0.63 | 0.01 |
| *Pseudomonas* | 920 | 3.12 | 0.05 | 904 | 2.76 | 0.05 |
| Eukaryota | 9333 | 31.65 | 0.47 | 9563 | 29.23 | 0.48 |
| *Nannochloropsis* | 5913 | 20.05 | 0.3 | 6179 | 18.88 | 0.31 |
| *N. limnetica* | 2223 | 7.54 | 0.11 | 2272 | 6.94 | 0.11 |
| Other | 1303 | 4.42 | 0.07 | 1284 | 3.92 | 0.06 |
| Total | 29,488 | 100 | 1.47 | 32,720 | 100 | 1.64 |

*N* number of identified sequences, *I* percentage of identified sequences, *A* percentage of all sequences included in the metagenomic analysis.

(Supplementary Fig. 6). We noted that the combined coverage was reduced across these two regions, with 21× and 32× chloroplast genome coverage (the latter calculated from 100 bp upstream and downstream of the deletion), which may be suggestive of within-sample variation. Both the reconstructed composite organellar genomes displayed authentic ancient DNA damage patterns (Supplementary Figs. 7 and 8).

To account for within-sample variants in our reconstructed organellar palaeogenomes, we created two consensus sequences that included either high- or low-frequency variants at multi-allelic sites. We performed phylogenetic analyses to confirm the placement of the high- and low-frequency variant consensus genomes relative to other *Nannochloropsis* taxa. For this, we used full organellar genomes and three short loci with high taxonomic representation in NCBI GenBank (18S, ITS, *rbc*L; Supplementary Data 4). Altogether, these analyses from three different markers (chloroplast, mitochondrial, nuclear) were congruent and resolved the high-frequency variant consensus sequences as likely deriving from *N. limnetica* var. *globosa* and the low-frequency variant consensus sequences as *N. limnetica* var. *limnetica* (Table 2, Fig. 4 and Supplementary Fig. 9).

We attempted to reconstruct composite chloroplast genomes using alternative *Nannochloropsis* taxa as seed sequences, but these analyses failed to resolve a complete composite sequence (Supplementary Table 3). A phylogenetic analysis of these alternative composite chloroplast genomes displays a topology consistent with the biases associated with mapping to increasingly diverged reference genomes (Supplementary Fig. 10). These alternative composite chloroplast genomes were therefore not used further, but provide supporting evidence that *N. limnetica* is the most closely related extant taxon.

**_Nannochloropsis limnetica_ allelic variation and haplotype estimation.** In the absence of a catalogue of chloroplast and mitochondrial genomes from the *N. limnetica* variants, we sought to explore the frequencies and proportions of allelic variants present in our dataset. We detected 299–376 and 81–112 transversion-only variants within the *N. limnetica* chloroplast and mitochondrial genomes, respectively (Supplementary Table 4). We could recover 64–70% of these transversion-only variants when variant calling was restricted to the damaged-only mapped datasets (Supplementary Table 4). For each sample and across the entire organellar genome, the average proportion of the

transversion-only alternative allele is 0.39–0.42 for chloroplast variants and 0.39–0.43 for mitochondrial variants (Fig. 5a, b).

After pooling data from both samples, we used the phasing of adjacent alleles, which were linked by the same read, to infer the minimum number of haplotypes in each reconstructed composite organellar genome. We identify 70 and 21 transversion-only phased positions in the chloroplast and mitochondrial genomes, respectively. Within each sample, the average number of haplotypes observed, based on the linked alleles in the chloroplast genome, is 1.93–2.09. The equivalent average for the mitochondrial genome is 2.05–2.29 (Fig. 5c and Supplementary Data 5).

**Metabarcoding analysis of Lake Øvre Æråsvatnet.** We generated 103,455,234 reads for the metabarcode libraries, with 38,112,900 (36.87%) retained after the bioinformatic analysis (Supplementary Methods). *Nannochloropsis* was detected in 93 out of the 192 samples (Supplementary Fig. 11 and Supplementary Data 6). Three different *Nannochloropsis* p6-loop sequences were detected in the metabarcode data. The two most abundant variants were detected in 70 and 83 samples and matched the p6-loop sequences derived from the high- and low-frequency variant chloroplast palaeogenomes, respectively (Supplementary Data 6 and Supplementary Table 5). The third variant was rarer and could only be detected in five samples, with only one PCR replicate detection out of eight for each of these samples.

**Meta-analysis of _Nannochloropsis_ in previous _sed_aDNA datasets.** *Nannochloropsis* could be detected in modern sedimentary DNA data from five north Norwegian localities[51] (Supplementary Table 6). In addition to LGM Andøya[41], *Nannochloropsis* has either been previously reported, or unreported, but present based on our re-analysis, in eight *sed*aDNA records from Greenland[52], St. Paul Island, Alaska, USA[9,11], Alberta and British Columbia, Canada[10], Latvia[53], Qinghai, China[54] and Svalbard[50,55] (Supplementary Data 7 and Supplementary Fig. 12).

**Discussion**
All of our analyses identified *Nannochloropsis* as the best represented identifiable eukaryotic taxon in the LGM lake sediments from Andøya, which is consistent with a previous plant DNA metabarcoding study of the same site[41]. These detections are comparable to other *sed*aDNA records, which suggest that *Nannochloropsis* is geographically and temporally widespread in lacustrine sediments (Supplementary Fig. 12). We also identified a *Mycobacterium* strain (or strains), but note that this is not closely related to any strains that have been sequenced to date and may therefore be extinct. The low relative frequencies of sequences mapping to other non-*Nannochloropsis* eukaryotic taxa are likely to be artifacts resulting from the spurious mapping of short and damaged ancient DNA molecules coupled with the vast diversity of sequences present in *sed*aDNA[22,23]. We note that the low overall proportions of sequences identified by our metagenomic analyses are broadly consistent with other shotgun metagenomic studies from *sed*aDNA[10,13,14] suggesting that the vast majority of taxonomic diversity in the sediment record is currently unidentifiable.

The detections of *Nannochloropsis* and *Mycobacterium* are not considered to be the result of contamination. The *sed*aDNA profiles of both taxa exhibit typical patterns of ancient DNA damage and are therefore considered to be of ancient origin. This is in contrast to the DNA fragments that were aligned to the human genome, which lacked such damage patterns, and are therefore considered to be of modern contaminant origin. In further support of the authenticity of our data, our reconstructed high and low-frequency *N. limnetica* organellar genomes both

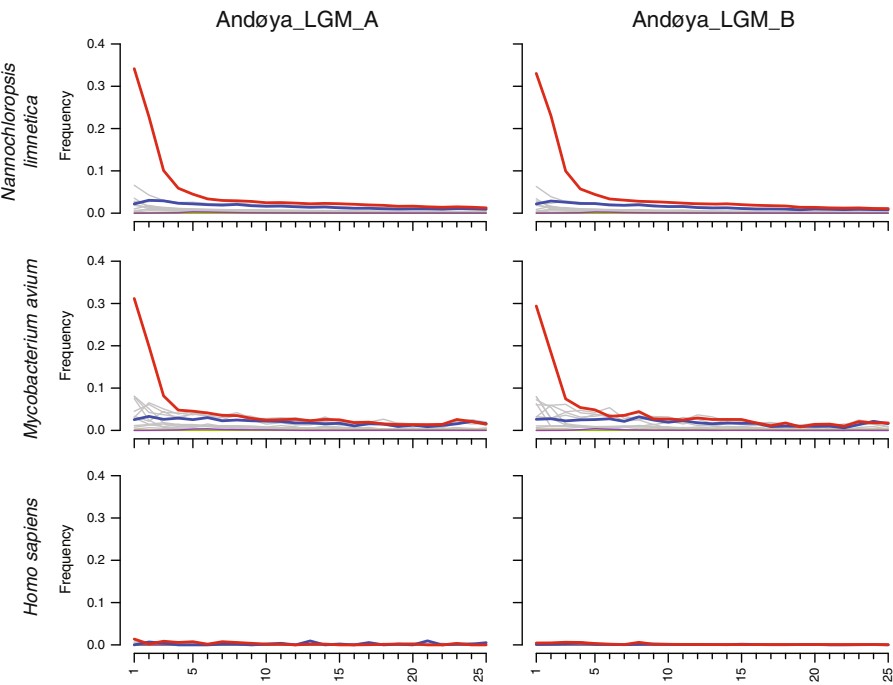

**Fig. 2 Ancient DNA deamination patterns for sequences aligned to the _Nannochloropsis limnetica_, _Mycobacterium avium_, and _Homo sapiens_ nuclear reference genomes.** Red lines are cytosine deamination profiles at the 5′ end of aligned sequences as calculated by mapDamage.

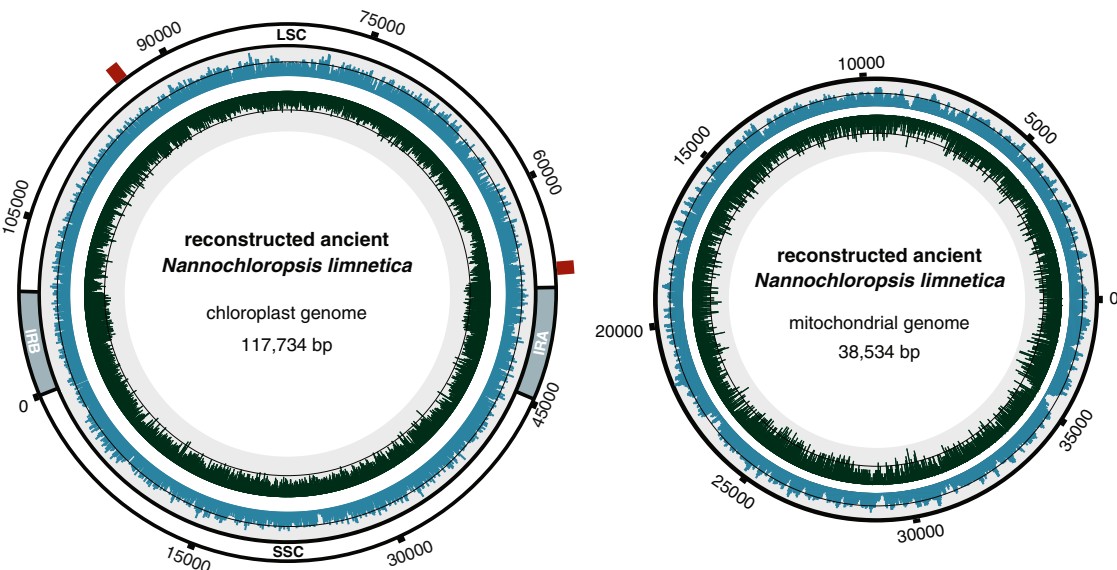

**Fig. 3 _Nannochloropsis limnetica_ chloroplast and mitochondrial palaeogenomes reconstructed directly from _seda_DNA.** The innermost circle contains a distribution of the GC content in dark green, with the black line representing the 50% mark. The outer blue distribution contains the genomic coverage for the assembly, with the black line representing the average coverage of 64.3× for the chloroplast and 64.9× for the mitochondria. For the chloroplast the inverted repeats (IRA and IRB), large single copy (LSC) and small single copy (SSC) regions are annotated. The red bars on the chloroplast indicate the location of the two regions with structural change compared to the _N. limnetica_ reference genome.

exhibit DNA damage, with the majority of transversion-only variants recovered in DNA damage-only alignments. This demonstrates that the observed variation is not driven by contaminating modern DNA. Furthermore, the reconstructed chloroplast variants can be distinguished on the basis of the _trn_L p6-loop locus, which are both detected in the same layers, and from the same DNA extracts, of this lake record using DNA metabarcoding (Supplementary Discussion and Supplementary

Fig. 11a). Importantly, neither of these variants were detected in any DNA extraction or PCR controls from these experiments (Supplementary Fig. 11b and Supplementary Data 6). Additionally, _Nannochloropsis_ was not detected in metabarcoding data from top sediments of the same lake[51], indicating that our observations are not the result of sampling contamination (Supplementary Table 6). Altogether, there is strong support for the authenticity of our _N. limnetica_ shotgun metagenomic data. In

**Table 2 Placement of the reconstructed high- and low-frequency organellar genomes and markers in each phylogenetic analysis.**

| Sample | Chloroplast Whole | | rbcL | | Mitochondrion Whole | | Nuclear 18S | | ITS | | Consensus |
|---|---|---|---|---|---|---|---|---|---|---|---|
| | Placement | BSS (%) | Placement | BSS (%) | Placement | BSS (%) | Placement | BSS (%) | Placement | BSS (%) | |
| Andøya_LGM_A HFV | Sister to N. limnetica | 100 | N. limnetica var. globosa | 57 | Sister to N. limnetica | 83 | N. limnetica | 76 | Sister to N. limnetica | 98 | N. limnetica var. globosa |
| Andøya_LGM_B HFV | Sister to N. limnetica | 100 | N. limnetica var. globosa | 57 | Sister to N. limnetica | 83 | N. limnetica | 76 | Sister to N. limnetica | 98 | N. limnetica var. globosa |
| Andøya_LGM_A LFV | Sister to N. limnetica | 100 | Sister to N. limnetica var. limnetica | 48 | Sister to N. limnetica | 83 | N. limnetica var. limnetica | 72 | Sister to N. limnetica | 49 | N. limnetica var. limnetica |
| Andøya_LGM_B LFV | Sister to N. limnetica | 100 | Sister to N. limnetica var. limnetica | 48 | Sister to N. limnetica | 83 | N. limnetica var. limnetica | 72 | Sister to N. limnetica | 49 | N. limnetica var. limnetica |

BSS bootstrap support values, HFV high-frequency variant, LFV low-frequency variant.

future larger-scale studies, it may be beneficial to leverage frequency-based methods for contaminant removal[56] in addition to sedaDNA damage and negative control evidence.

We explored whether Andøya Nannochloropsis could potentially comprise more than one species, but found no evidence to suggest this was the case. The phylogenetic placement of our organellar palaeogenomes, as well as other short loci, indicate that two varieties of N. limnetica were present, with both recovered at similar frequencies from our broadly contemporaneous samples, thereby demonstrating the replicability of our results. Our method based on the phasing of adjacent variable positions suggests the presence of at least three haplotypes. We note that this method is likely to have been limited by the short fragment lengths characteristic of sedaDNA, which may explain the lower proportion of three-haplotype counts for Andøya_LGM_B that had a shorter mean fragment length. Further resolution of haplotype diversity was hampered by the lack of an extensive genomic reference database of N. limnetica haplotypes, which may be required for true haplotype estimation. However, this would be particularly problematic for taxa that lack the body fossils currently required to reconstruct extinct haplotypes. Future methodological and statistical advances will therefore be required to estimate and quantify true haplotype variation for taxa from within a sedaDNA population sample.

Our complete N. limnetica chloroplast palaeogenome reconstructions represent the first derived from sedaDNA to the best of our knowledge, although a near-complete chloroplast sequence has recently been reported for a vascular plant[14]. Mitochondrial palaeogenomes have previously been reconstructed from cave sediments[12], and archaeological middens and latrines[24,25], but ours are, as far as we know, the first derived from lake sediments. The high depth of coverage for our sample-combined palaeogenomes (62–64×) allowed us to explore allelic proportions and haplotype diversity using sedaDNA. We identified two distinct haplogroups, with our method indicating that at least three haplotypes were present. While advances in obtaining population genomic information from modern multi-taxon DNA datasets have been made[16–21], the application of these methods to sedaDNA is non-trivial. The method presented here was able to uncover more haplotype information from degraded multi-taxon DNA than would have been possible by analysis of single variable sites. Together with recently published and ongoing studies, our work demonstrates the feasibility of the sedaDNA field moving into a next phase of environmental palaeogenomics. This will enable a broad range of ecological and evolutionary questions to be addressed using population genomic approaches, including for communities of taxa that may or may not be preserved in the body fossil record. With further innovations, this approach could also be extended to a suite of broad groups, including plants, invertebrates, and vertebrates, from lake catchments, cave sediments, and archaeological settings, therefore unlocking the full potential of sedaDNA.

## Methods

**Site description, chronology, and sampling**. A detailed description of the site, coring methods, age-depth model reconstruction, and sampling strategy can be found in Alsos et al.[41]. Briefly, Lake Øvre Ærråsvatnet is located on Andøya, Northern Norway (69.25579°N, 16.03517°E) (Fig. 1a, b). In 2013, two cores were collected from the deepest sediments, AND10 and AND11, which were stored at 4 °C prior to sampling. Macrofossil remains were dated, with those from AND10 all dating to within the LGM. For the longer core AND11, a Bayesian age-depth model was required to estimate the age of each layer[41]. In this study, we selected one sample of LGM sediments from each of the two cores. According to the Bayesian age-depth model, sample Andøya_LGM_B, from 1102 cm depth in AND11, was dated to a median age of 17,700 (range: 20,200–16,500) cal yr BP. The age of Andøya_LGM_A, from 938 cm depth in AND10, was estimated at 19,500 cal yr BP, based on the interpolated median date between two adjacent macrofossils (20 cm above: 19,940–18,980 cal yr BP, 30 cm below: 20,040–19,000 cal yr BP). As

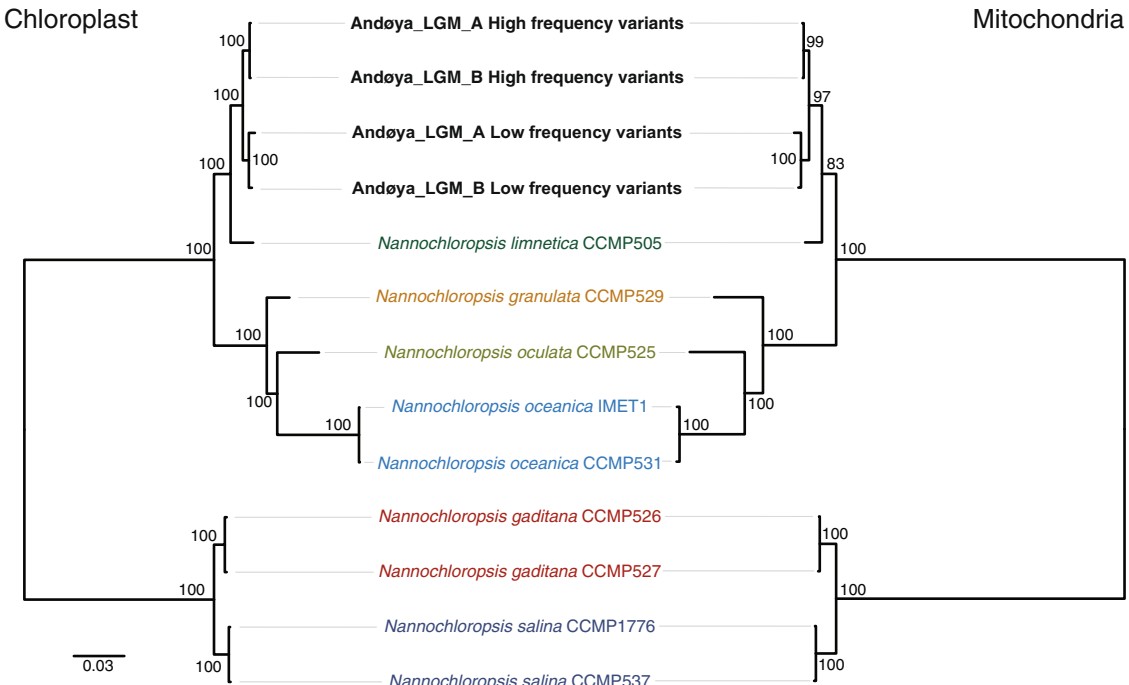

**Fig. 4 Maximum likelihood phylogenies of *Nannochloropsis* organellar genomes.** Maximum likelihood phylogenies of *Nannochloropsis* chloroplast (left) and mitochondrial (right) genome sequences, including the reconstructed *N. limnetica* consensus sequences based on either high- or low-frequency variants. Node values indicate the bootstrap support.

Andøya_LGM_A falls within the age range of Andøya_LGM_B, we consider the samples to be broadly contemporaneous.

**Sampling, DNA extraction, library preparation, and sequencing**. The two cores were subsampled at the selected layers under clean conditions, in a dedicated ancient DNA laboratory at The Arctic University Museum of Norway in Tromsø. We extracted DNA from 15 g of sediment following the Taberlet phosphate extraction protocol[29] in the same laboratory. We shipped a 210 μL aliquot of each DNA extract to the ancient DNA dedicated laboratories at the Centre for GeoGenetics (University of Copenhagen, Denmark) for double-stranded DNA library construction. The extracts were concentrated to 80 μL using Amicon Ultra-15 30 kDa centrifugal filters (Merck Millipore, Darmstadt, Germany) and half of each extract (40 μL, totalling between 31.7 and 36.0 ng of DNA) was converted into Illumina-compatible libraries using established protocols[10]. Each library was dual-indexed via 12 cycles of PCR. The libraries were then purified using the AmpureBead protocol (Beckman Coulter, Indianapolis, IN, USA), adjusting the volume ratio to 1:1.8 library:AmpureBeads, and quantified using a BioAnalyzer (Agilent, Santa Clara, CA, USA). The indexed libraries were pooled equimolarly and sequenced on a lane of the Illumina HiSeq 2500 platform using 2 × 80 cycle paired-end chemistry.

**Raw read filtering**. For each sample, we merged and adapter-trimmed the paired-end reads with *SeqPrep* (https://github.com/jstjohn/SeqPrep/releases, v1.2) using default parameters. We only retained the resulting merged sequences, which were then filtered with the preprocess function of the *SGA toolkit* v0.10.15 (ref. [57]) by the removal of those shorter than 35 bp or with a DUST complexity score > 1.

**Metagenomic analysis of the sequence data**. We first sought to obtain an overview of the taxonomic composition of the samples and therefore carried out a BLAST-based metagenomic analysis on the two filtered sequence datasets. To make the datasets more computationally manageable, we subsampled the first and last one-million sequences from the filtered dataset of each sample and analysed each separately. The data subsets were each identified against the NCBI nucleotide database (release 223) using the *blastn* function from the *NCBI-BLAST+* suite v2.2.18+[58] under default settings. For each sample, the results from the two subsets were checked for internal consistency, merged into one dataset, and loaded into *MEGAN* v6.12.3 (ref. [59]). Analysis and visualization of the Last Common Ancestor (LCA) was carried out for the taxonomic profile using the following settings: min score = 35, max expected = 1.0E−5, min percent identity = 95%, top percent = 10%, min support percentage = 0.01, LCA = naive, min percent sequence to cover = 95%. We define sequences as the reads with BLAST hits assigned to taxa post-filtering, thus ignoring "unassigned" and "no hit" categories.

**Alignment to reference genome panels**. We mapped our filtered data against three different reference panels to help improve taxonomic identifications and provide insight into the sequence abundance of the identified taxa (Supplementary Data 2 and 3). The first reference panel consisted of 42 nuclear genomes that included genera expected from Northern Norway, exotic/implausible taxa for LGM Andøya, human, six *Nannochloropsis* species, and four strains of *Mycobacterium*. The inclusion of exotic taxa was to give an indication of the background spurious mapping rate, which can result from mappings to conserved parts of the genome and/or short and damaged ancient DNA molecules[22,23]. We included *Nannochloropsis*, *Mycobacterium*, and human genomes, due to their overrepresentation in the BLAST-based metagenomic analysis. The other two reference panels were based on either all 8486 mitochondrial or 2495 chloroplast genomes on NCBI GenBank (as of January 2018). The chloroplast dataset was augmented with 247 partial or complete chloroplast genomes generated by the PhyloNorway project[60] for 2742 chloroplast genomes in total. The filtered data were mapped against each reference genome or organellar genome set individually using *bowtie2* v2.3.4.1 (ref. [61]) under default settings. The resulting bam files were processed with *SAMtools* v0.1.19 (ref. [62]). We removed unmapped sequences with *SAMtools view* and collapsed PCR duplicate sequences with *SAMtools rmdup*.

For the nuclear reference panel, we reduced potential spurious or nonspecific sequence mappings by comparing the mapped sequences to both the aligned reference genome and the NCBI nucleotide database using *NCBI-BLAST+*, following the method used by Graham et al.[9], as modified by Wang et al.[11]. The sequences were aligned using the following *NCBI-BLAST+* settings: num_alignments = 100 and perc_identity = 90. Sequences were retained if they had better alignments, based on bit score, to reference genomes as compared to the NCBI nucleotide database. If a sequence had a better or equal match against the NCBI nucleotide database, it was removed, unless the LCA of the highest NCBI nucleotide bit score was from the same genus as the reference genome (based on the NCBI taxonID). To standardize the relative mapping frequencies to genomes of different size, we calculated the number of retained mapped sequences per Mb of genome sequence.

The sequences mapped against the chloroplast and mitochondrial reference panels were filtered and reported in a different manner than the nuclear genomes. First, to exclude any non-eukaryotic sequences, we used *NCBI-BLAST+* to search sequence taxonomies and retained sequences if the LCA was, or was within, Eukaryota. Second, for the sequences that were retained, the LCA was calculated and reported in order to summarize the mapping results across the organelle datasets. LCAs were chosen as the reference sets are composed of multiple genera.

Within the *Nannochloropsis* nuclear reference alignments, the relative mapping frequency was highest for *N. limnetica*. In addition, the relative mapping frequency for other *Nannochloropsis* taxa was higher than those observed for the exotic taxa. This could represent the mapping of sequences that are conserved between *Nannochloropsis* genomes or suggest the presence of multiple *Nannochloropsis* taxa in a community sample. We therefore cross-compared mapped sequences to determine the number of uniquely mapped sequences per reference genome. First,

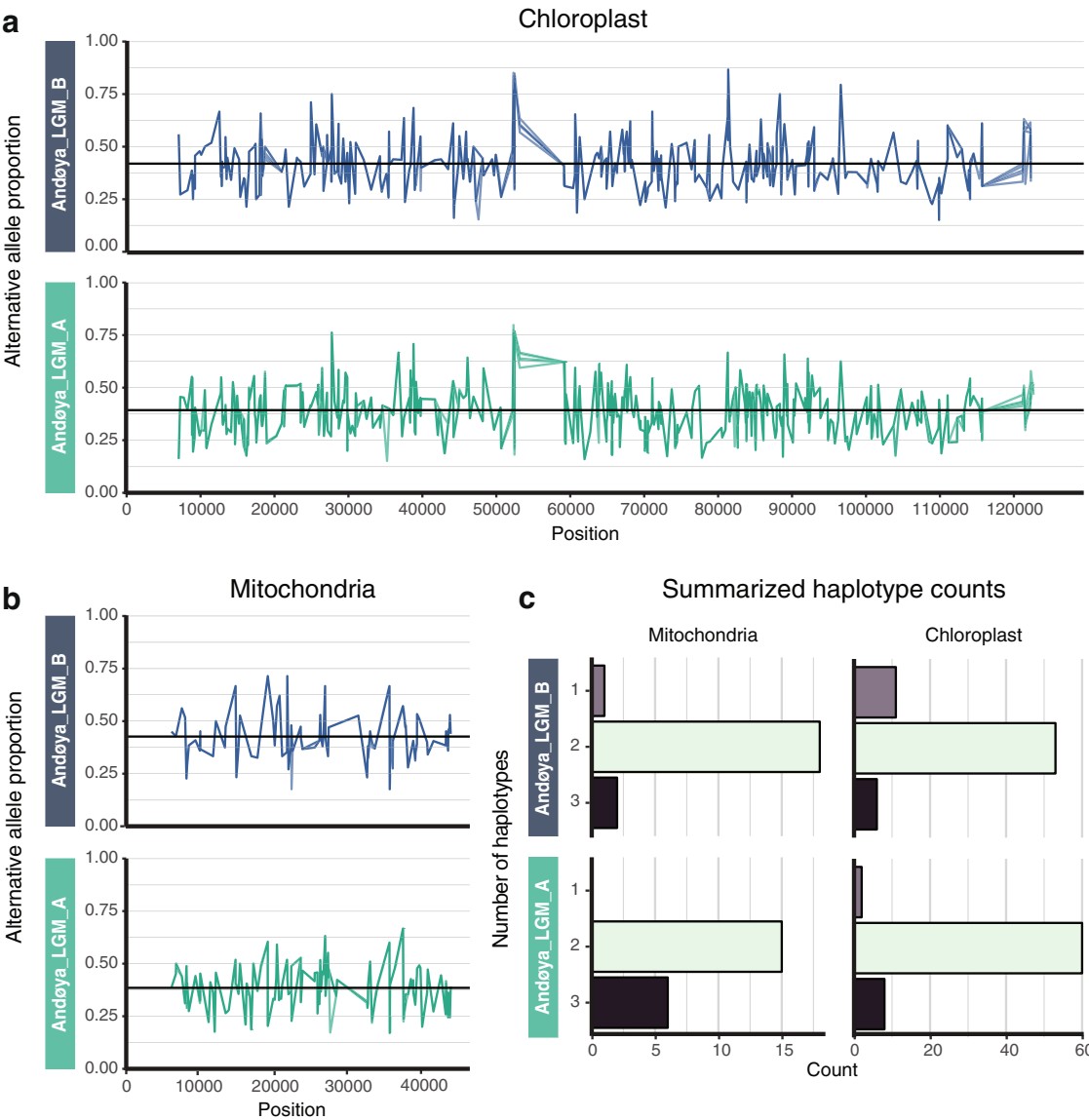

**Fig. 5 The proportions of variant sites and minimum haplotype counts for *Nannochloropsis* organellar genomes.** The proportions of alternative alleles across the chloroplast (**a**) and mitochondrial (**b**) genomes based on transversions-only. Each proportion plot consists of five independent variant calling runs to account for sampling biases (see "Methods"). The horizontal black lines represent averages: 0.39 and 0.42 for the chloroplast and 0.39 and 0.43 for the mitochondria, for samples Andøya_LGM_A and Andøya_LGM_B, respectively. In **a** and **b**, colour denotes sample. **c** Observed minimum haplotype counts based on the linked alleles for the chloroplast and mitochondrial genomes for both samples.

we individually remapped the filtered data to six available *Nannochloropsis* nuclear genomes, the accession codes of which are provided in Supplementary Data 2. For each sample, we then calculated the number of sequences that uniquely mapped to, or overlapped, between each *Nannochloropsis* genome. We repeated the above analysis with six available chloroplast sequences (Supplementary Data 2) to get a comparable overlap estimation for the chloroplast genome.

**Reconstruction of the Andøya *Nannochloropsis* community organellar palaeogenomes.** To place the Andøya *Nannochloropsis* community taxon into a phylogenetic context, and provide suitable reference sequences for variant calling, we reconstructed environmental palaeogenomes for the *Nannochloropsis* mitochondria and chloroplast. First, the raw read data from both samples were combined into a single dataset and re-filtered with the *SGA toolkit* to remove sequences shorter than 35 bp, but retain low complexity sequences to assist in the reconstruction of low complexity regions in the organellar genomes. This re-filtered sequence dataset was used throughout the various steps for environmental palaeogenome reconstruction.

The re-filtered sequence data were mapped onto the *N. limnetica* reference chloroplast genome (NCBI GenBank accession: NC_022262.1) with *bowtie2* using default settings. *SAMtools* was used to remove unmapped sequences and PCR

duplicates, as above. We generated an initial consensus genome from the resulting bam file with *BCFtools* v1.9 (ref. [62]), using the *mpileup*, *call*, *filter*, and *consensus* functions. For variable sites, we produced a majority-rule consensus using the --variants-only and --multiallelic-caller options, and for uncovered sites the reference genome base was called. The above steps were repeated until the consensus could no longer be improved. The re-filtered sequence data was then remapped onto the initial consensus genome sequence with *bowtie2*, using the above settings. The *genomecov* function from *BEDtools* v2.17.0 (ref. [63]) was used to identify gaps and low coverage regions in the resulting alignment.

We attempted to fill the identified gaps, which likely consisted of diverged or difficult-to-assemble regions. For this, we assembled the re-filtered sequence dataset into de novo contigs with the MEGAHIT pipeline v1.1.4 (ref. [64]), using a minimum *k*-mer length of 21, a maximum *k*-mer length of 63, and *k*-mer length increments of six. The MEGAHIT contigs were then mapped onto the initial consensus genome sequence with the *blastn* tool from the *NCBI-BLAST+* toolkit. Contigs that covered the gaps identified by *BEDtools* were incorporated into the initial consensus genome sequence, unless a *blast* comparison against the NCBI nucleotide database suggested a closer match to non-*Nannochloropsis* taxa. We repeated the *bowtie2* gap-filling steps iteratively, using the previous consensus sequence as reference, until a gap-free consensus was obtained. The re-filtered sequence data were again mapped, the resulting final assembly was visually

inspected, and the consensus was corrected where necessary. This was to ensure the fidelity of the consensus sequence, which incorporated de novo-assembled contigs that could potentially be problematic, due to the fragmented nature and deaminated sites of ancient DNA impeding accurate assembly[65].

Annotation of the chloroplast genome was carried out with *GeSeq* v1.77 (ref. [66]), using the available annotated *Nannochloropsis* chloroplast genomes (accession codes provided in Supplementary Table 7). The resulting annotated chloroplast was visualized with *OGDRAW* v1.3.1 (ref. [67]).

The same assembly and annotation methods outlined above were used to reconstruct the mitochondrial palaeogenome sequence, where the initial mapping assembly was based on the *N. limnetica* mitochondrial sequence (NCBI GenBank accession: NC_022256.1). The final annotation was carried out by comparison against all available annotated *Nannochloropsis* mitochondrial genomes (accession codes provided in Supplementary Table 7).

If the *Nannochloropsis* sequences derived from more than one taxon, then alignment to the *N. limnetica* chloroplast genome could introduce reference bias, which would underestimate the diversity of the *Nannochloropsis* sequences present. We therefore reconstructed *Nannochloropsis* chloroplast genomes, but using the six available *Nannochloropsis* chloroplast genome sequences, including *N. limnetica*, as seed genomes (accession codes for the reference genomes are provided in Supplementary Table 3). The assembly of the consensus sequences followed the same method outlined above, but with two modifications to account for the mapping rate being too low for complete genome reconstruction based on alignment to the non-*N. limnetica* reference sequences. First, consensus sequences were called with *SAMtools*, which does not incorporate reference bases into the consensus at uncovered sites. Second, neither additional gap filling nor manual curation was implemented.

### Analysis of ancient DNA damage patterns.
We checked for the presence of characteristic ancient DNA damage patterns for sequences aligned to three nuclear genomes: human, *Nannochloropsis limnetica* and *Mycobacterium avium*. We further analysed damage patterns for sequences aligned to both the reconstructed *N. limnetica* composite organellar genomes. Damage analysis was conducted with *mapDamage* v2.0.8 (ref. [68]) using the following settings: --merge-reference-sequences and --length = 160.

### Assembly of high- and low-frequency variant consensus sequences.
The within-sample variants in each reconstructed organellar palaeogenome was explored by creating two consensus sequences, which included either high- or low-frequency variants at multiallelic sites. For each sample, the initial filtered sequence data were mapped onto the reconstructed *Nannochloropsis* chloroplast palaeogenome sequence with *bowtie2* using default settings. Unmapped and duplicate sequences were removed with *SAMtools*, as above. We used the *BCFtools mpileup*, *call*, and *normalize* functions to identify the variant sites in the mapped dataset, using the --skip-indels, --variants-only, and --multiallelic-caller options. The resulting alleles were divided into two sets, based on either high- or low-frequency variants. High-frequency variants were defined as those present in the reconstructed reference genome sequence. Both sets were further filtered to only include sites with a quality score of 30 or higher and a coverage of at least half the average coverage of the mapping assembly (minimum coverage: Andøya_LGM_A = 22×, Andøya_LGM_B = 14×). We then generated the high- and low-frequency variant consensus sequences using the consensus function in *BCFTools*. The above method was repeated for the reconstructed *Nannochloropsis* mitochondrial genome sequence in order to generate comparable consensus sequences of high- and low-frequency variants (minimum coverage: Andøya_LGM_A = 16×, Andøya_LGM_B = 10×).

### Phylogenetic analysis of the reconstructed organellar palaeogenomes.
We determined the phylogenetic placement of our high- and low-frequency variant organellar palaeogenomes within *Nannochloropsis*, using either full mitochondrial and chloroplast genome sequences or three short loci (18S, ITS, *rbc*L). We reconstructed the 18S and ITS1-5.8S-ITS2 complex using DQ977726.1 (full length) and EU165325.1 (positions 147:1006, corresponding to the ITS complex) as seed sequences following the same approach that was used for the organellar palaeo-genome reconstructions, except that the first and last 10 bp were trimmed to account for the lower coverage due to sequence tiling. We then called high and low variant consensus sequences as described above.

We created six alignments using available sequence data from NCBI GenBank (Supplementary Data 4) with the addition of: (1 + 2) the high- and low-frequency variant chloroplast or mitochondrial genome consensus sequences, (3) an ~1100 bp subset of the chloroplast genome for the *rbc*L alignment, (4 + 5) ~1800 and ~860 bp subsets of the nuclear multicopy complex for the 18S and ITS alignments, respectively, and (6) the reconstructed chloroplast genome consensus sequences derived from the alternative *Nannochloropsis* genome starting points. Full details on the coordinates of the subsets are provided in Supplementary Data 4. We generated alignments using *MAFFT* v7.427 (ref. [69]) with the maxiterate = 1000 setting, which was used for the construction of a maximum likelihood tree in *RAxML* v8.1.12 (ref. [70]) using the GTRGAMMA model and without outgroup specified. We assessed branch support using 1000 replicates of rapid bootstrapping.

### *Nannochloropsis* variant proportions and haplogroup diversity estimation.
To estimate major haplogroup diversity, we calculated the proportions of high and low variants in the sequences aligned to our reconstructed *Nannochloropsis* mitochondrial and chloroplast genomes. For each sample, we first mapped the initial filtered sequence data onto the high- and low-frequency variant consensus sequences with *bowtie2*. To avoid potential reference biases, and for each organellar genome, the sequence data were mapped separately against both frequency consensus sequences. The resulting bam files were then merged with *SAMtools merge*. We removed exact sequence duplicates, which may have been mapped to different coordinates, from the merged bam file by randomly retaining one copy. This step was replicated five times to examine its impact on the estimated variant proportions. After filtering, remaining duplicate sequences—those with identical mapping coordinates—were removed with *SAMtools rmdup*. We then called variable sites from the duplicate-removed bam files using *BCFTools* under the same settings as used in the assembly of the high- and low-frequency variant consensus sequences. We restricted our analyses to transversion-only variable positions to remove the impact of ancient DNA deamination artifacts. For each variable site, the proportion of reference and alternative alleles was calculated, based on comparison to the composite *N. limnetica* reconstructed organellar palaeogenomes. We removed rare alleles occurring at a proportion of <0.1, as these may have resulted from noise.

To ensure that the variants detected are not driven by contaminating modern DNA, we repeated the above variant calling process using a restricted dataset that only included damaged sequences. We used *PMDtools* v0.60 (ref. [71]) to extract sequences from the above high- and low-frequency bam files for each sample and organellar genome with a DNA damage score threshold of ≥3. We then repeated the *SAMtools* merge and variant calling process, as described above, to confirm that the variants identified could be recovered based on reads exhibiting ancient DNA damage. The variant calling was altered to use a minimum SNP quality threshold of three and minimum coverage of three, in order to recover calls from the lower coverage dataset.

To infer the minimum number of haplogroups in each reconstructed organellar genome sequence, we inspected the phasing of adjacent variable sites that were linked by the same read in the duplicate-removed bam files, akin to the method used by Søe et al.[25] (Supplementary Fig. 13). For this, we first identified all positions, from both samples, where two or more transversion-only variable sites occurred within 35 bp windows. For each window, we then examined the allelic states in mapped sequences that fully covered the linked positions. We recorded the combination and frequency of alleles for each window of linked positions to calculate the observed haplotype diversity. We removed low-frequency haplotypes, which were defined as those with <3 sequences or <15% of all sequences that covered a linked position, and the remaining haplotypes were scored. These steps were repeated for each set of linked positions across both organellar genomes for both of the metagenomic datasets.

**Reporting summary.** Further information on research design is available in the Nature Research Reporting Summary linked to this article.

## Data availability
The raw Illumina shotgun metagenomic datasets, as well as the new and reanalysed metabarcoding data from Lake Øvre Æråsvatnet and Alsos et al.[55], are available from the European Nucleotide Archive (ENA) under accession PRJEB38213. The reconstructed *Nannochloropsis limnetica* high- and low-frequency organellar genome sequences are available from NCBI GenBank under accessions MT872223-MT872230.

## Code availability
The script for estimating the number of haplotypes across the linked windows has been uploaded to GitHub at https://github.com/Y-Lammers/HaplogroupEstimation (v1.0)[72].

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

## Acknowledgements

This paper is a part of a larger project on the past environment of Andøya and we thank the Andøya team for fruitful discussion and access to pre-publication results. We thank Per Sjögren, Aage Paus and Ludovic Gielly for assistance with fieldwork; Antony G. Brown for help with the age-depth models; Mikkel W. Pedersen for conducting the library preparation and overseeing the sequencing; Edana Lord, Vendela K. Lagerholm and Love Dalén for access to the pre-published *Lemmus lemmus* genome; and Sandra Garcés Pastor for informative discussions. We also thank Pierre Taberlet for constructive feedback on our manuscript. The work was funded by the Research Council of Norway (grants: 213692, Ancient DNA of NW Europe reveals responses of climate change; 250963, ECOGEN—Ecosystems change and species persistence over time: a genome-based approach to IGA). Y.L. was financed by an internal PhD position at The Arctic University Museum of Norway.

## Author contributions

Y.L., P.D.H., and I.G.A. conceptualized and designed the research, and contributed to the final version of the manuscript; Y.L. analysed the data and wrote the first draft; P.D.H. provided analytical guidance and refined the drafted manuscript; I.G.A. performed fieldwork, DNA extractions, provided resources, acquired funding, and supervised Y.L.

## Competing interests

The authors declare no competing interests.
