## [Peer Review File · Communications Biology]

Reviewers' comments:

Reviewer #1 (Remarks to the Author):

This paper is an excellent study on the potential of environmental palaeogenomics. The article investigates the feasibility of shotgun sequencing lake sediment samples to obtain population genomic information for taxa that do not preserve in the body fossil record. With a solid bioinformatic approach, authors were able to profile taxa and get population genomic information (organelle data) from the most abundant taxa, *Nannochloropsis limnetica*. The conclusions are well supported, and the findings are of general interest, especially because environmental palaeogenomics is an active area of research in the eDNA field. I enjoyed this study, and I don't have substantive comments, just a few suggestions and a minor correction that are only given for authors' consideration/improvement and by no means decrease the value of the paper:

General comments:

- Possibility of contamination: The authors did a great job demonstrating that the *N. limnetica* sequencing data exhibited ancient DNA characteristics, and thus, I find it very unlikely that their results are explained by contamination. Despite that, I wonder if there is any possibility that the sample was contaminated with contemporary *N. limnetica* DNA during collection. In particular, it is unclear to me if this taxon is currently present in this lake. And, I couldn't access Alsos et al., 2020 (31) to get more details on the sampling procedures (because I think it is not yet available). If the taxon is currently present, I recommend the authors elaborate a bit further on why –in addition to the evidence of ancient DNA damage patterns– they consider there is no contamination from contemporary *N. limnetica*. Also, considering that future studies are likely to follow the bioinformatic approach of this study, I would recommend the authors consider including a brief statement on the use of decontamination methods. For instance, for studies with larger sample size, it may be possible to use Decontam (Davis et al., 2018) (frequency method) over the metagenomic OTU table to see if any taxa exhibit the property of being a contaminant (i.e., an inverse relationship with total DNA amount). This way, taxa suspicious of being a contaminant could be excluded from further analyses (e.g., pop genomics).
- Third-generation sequencing: In line with what I wrote above, I suppose long reads could improve the quality of the data that could be obtained, despite how damaged the DNA is (e.g., long organelle contigs). Accordingly, I would also briefly discuss this point on what else needs to be explored (e.g., among 200-206 lines).

Minor comments:

- Line 118: Supplementary Figures S3 and S4.
- Lines 469-471: I would recommend making this easier to understand.
- The version of some bioinformatic software tools is missing in the main text, e.g., OGDRAW

Reviewer #2 (Remarks to the Author):

Overview: The study utilized two sediment cores to assess the presence of an algae genera in a lake at a single time point using modern molecular methods. The take home message is that their findings verifies the use of molecular approaches on sediment cores to assess historical community dynamics. While the molecular approach seems fine, the low sampling power and how the issue of sediment

mixing should be addressed. The writing could also be improved as the paper is very organism specific at the moment and will have limited relevance to a wider audience. The bioinformatics pipeline to discern haplotypes itself holds some merit and may be a better focus instead of trying to fit the study into a palaeological context.

Line 64-67: This is overly specific, there are several studies that have used molecular tools on non-targeted DNA to derive genomic diversity. The introduction should include a broader research focus to include the methods used.

Methods: There should be some text on how the possibility of false positives were taken into account with regards to the study design. The main issue being that taking a single DNA sample from each of the two cores leave a lot of room for random error in any downstream analyses.

Results: A lot of methods text is present that distracts from the results. I understand the format is difficult, but the section needs to be revised to focus on results and not methodology. Presently, it is not clear what the results were. Additionally, several parts are interpretative and should be moved to the discussion. E.g. lines 93-104, 106-110, 111-120, etc....

Discussion: I would suggest focusing on the haplotype diversity analyses. The abundance/community interpretations are not well supported given the sampling design.

Line 207-222: There is no statistical variation about the sampling points to infer differences in abundances of the current data set. The high abundance is just as likely to be a random sampling error.

Line 234-237: Again, there are several existing studies using similar methods.

Reviewer #3 (Remarks to the Author):

Dear authors,

The manuscript „Environmental reconstruction of an Ice Age algal population“ authored by Youri Lammers, Peter Heintzmann and Inger Alsos presents a study about the identification of chloroplast and mitochondrial genomes of the algae *Nannochloropsis* detected in shotgun (metagenomic) data from two late glacial sediment samples from a lake in Northern Norway. The manuscript is well-written and technically convincing. The application of metagenomics on ancient lake sedimentary DNA is still a relatively novel approach, with only a few studies published so far. This study adds new knowledge about late glacial taxonomic composition in a northern lake, which seems largely restricted to a few, but dominant taxa during this time period.

General comments:

The study gives support for a dominance of algae growth during the late glacial because the majority

of classified reads are assigned to *Nannochloropsis*. The authors applied a robust taxonomic assignment which supports the authenticity of detected ancient DNA reads. The authors also point out that two different samples originating from a similar time slice, but from different sediment cores are highly similar in terms of taxonomic assignment. So far, the manuscript focuses on taxonomic classification and the identification of haplotypes from metagenomic data and these analyses seem very convincing. Beyond this, I am wondering if the authors could add additional information to the main manuscript, which is:

- provide an overview (e.g. pie chart for sample A and B) for unidentified and classified reads (incl. tax. assignment) - this would give a nice overview and one does not need the Supplementary tables to understand the data basis
- indicate which parts of the chloroplast or mitochondrial genome of *Nannochloropsis* are possibly overrepresented in the ancient reads or are not covered at all (did you see an effect of PCR duplicates, which might have happened during library amplification in the indexing PCR?)
- is it possible to say if mitochondrial or chloroplast DNA is better preserved? What is known about the abundance of mitochondria and chloroplasts in *Nannochloropsis*?
- Does your data support also evidence for nuclear *Nannochloropsis* DNA?
- Are barcode regions (like ITS, 18S, rbcL) represented in the ancient reads? Could one use the metagenomic data and compare with metabarcoding?
- Sample A seems to perform better than B, because it shows an overall higher read count and longer mean read length, does this affect the taxonomic assignment of the data? What might be the reason for this difference?

It is interesting that the authors detected *Mycobacterium avium* in their data. Could there be any connection between the adjacent bird cliff and the occurrence of *mycobacterium avium* in the sediments? It is also known that *Mycobacterium* can produce resting cells (see Wu et al. Developmental transcriptome of resting cell formation in *Mycobacterium smegmatis*) and survive in soils for many days (Tio-Coma et al. "Detection of *Mycobacterium leprae* DNA in soil: multiple needles in the haystack"). Maybe these features enhance DNA preservation over time?

Although the authors show typical ancient patterns in the *Nannochloropsis* and *Mycobacterium* reads, I completely miss the investigation of extraction blanks and library controls. Please explain how you used controls during all preparation steps (from DNA extraction to final sequencing reads) to monitor possible contamination sources. If you also sequenced negative controls, I would highly recommend to present the results in the manuscript.

Detailed comments:

The authors might think about to change the very general title of the manuscript in a more specific one, because the promised "environmental reconstruction" is only a very little part in the manuscript. Maybe you use a title like: "Nannochloropsis organelle genomes retrieved from a metagenomic analyses of glacial lake sediments in Northern Norway".

Line 242

Please indicate shortly if the core was stored frozen or at 4°C.

Line 262

How did you concentrate the sample? Did you use a kit? Please specify.

Line265

How many PCR cycles were used in the indexing PCRs?

Line 269

2x80 cycles, you might lose DNA fragments which are very long, and can't merged after sequencing? Do you think this might affect your data? Please specify the average fragment length of your libraries

Line 275

Please specify the number of unmerged reads with good quality. Give the relative abundance of merged and unmerged sequences based on the raw read count.

Line 703

Please provide the unit of the y-axis in Figure 1c. I guess for algae and bacteria it is total reads and for human and others % of filtered reads?

Line 221

Supplementary Figure 11 is cited, but not present in the Supplement. Should be 9, I guess.

Supplementary Figure S9: Overview of Nannochloropsis detections, using 10 sedaDNA...

Please indicate if these results were retrieved by metabarcoding or metagenomic data, and if metabarcoding was used, please indicate which metabarcode has been applied. For example, the metabarcoding approach using the P6 loop (Taberlet et al. 2007) is not designed for the detection of algae in sediments, which will likely produce false negatives. Typically, the amplicon for Nannochloropsis using the P6 loop metabarcode is very short (about 12bp) and shows low identities to the reference in the databases. For this reason, I am not sure, if the detection obtained from non-specific metabarcoding will help to conclude on a distribution pattern through time. I am also not sure, what is the conclusion out of this compilation for the shot gun data?

Reviewers' comments:

Reviewer #1 (Remarks to the Author):

This paper is an excellent study on the potential of environmental palaeogenomics. The article investigates the feasibility of shotgun sequencing lake sediment samples to obtain population genomic information for taxa that do not preserve in the body fossil record. With a solid bioinformatic approach, authors were able to profile taxa and get population genomic information (organelle data) from the most abundant taxa, *Nannochloropsis limnetica*. The conclusions are well supported, and the findings are of general interest, especially because environmental palaeogenomics is an active area of research in the eDNA field. I enjoyed this study, and I don't have substantive comments, just a few suggestions and a minor correction that are only given for authors' consideration/improvement and by no means decrease the value of the paper

We thank the reviewer for their kind comments.

General comments:

R1.1 • Possibility of contamination: The authors did a great job demonstrating that the *N. limnetica* sequencing data exhibited ancient DNA characteristics, and thus, I find it very unlikely that their results are explained by contamination. Despite that, I wonder if there is any possibility that the sample was contaminated with contemporary *N. limnetica* DNA during collection.

In particular, it is unclear to me if this taxon is currently present in this lake.

R1.1 response. There are several lines of evidence to suggest that our results are not affected by contemporary *N. limnetica* contamination.

First, *N. limnetica* is unlikely to be a laboratory contaminant, as we do not detect the shotgun-derived *Andøya Nannochloropsis* barcode sequences in the metabarcoding controls for both this study or Alsos *et al.* 2020². Second, as *N. limnetica* is a photosynthetic microalgae, we consider the possibility that it would be free-living in deep LGM sediments to be highly improbable. A more likely source of modern contamination would be during core

collection and potential contact with lake water. However, plant metabarcoding results from a study of modern, top sediment from the same lake did not detect *Nannochloropsis*¹ (Supplementary Table S12). We also took great care during *seda*DNA sampling of the core by excluding sediment that had previously been in contact with the atmosphere and coring pipe, to avoid this type of contamination².

We have updated the discussion (L.190-205) with a paragraph discussing the issue of potential contamination, which covers the above points that our results are highly unlikely to originate from modern contamination.

R1.2 • And, I couldn't access Alsos et al., 2020 (31) to get more details on the sampling procedures (because I think it is not yet available).

R1.2 response. We were hoping that Alsos *et al.* 2020² would be available by the time of review of this manuscript, but this was unfortunately not the case. However, Alsos *et al.* (2020) is now published and the sampling procedures are available for scrutiny (doi: 10.1016/j.quascirev.2020.106364).

R1.3 • If the taxon is currently present, I recommend the authors elaborate a bit further on why – in addition to the evidence of ancient DNA damage patterns– they consider there is no contamination from contemporary *N. limnetica*.

R1.3 response. We have addressed this concern in our response to point R1.1 above.

R1.4 • Also, considering that future studies are likely to follow the bioinformatic approach of this study, I would recommend the authors consider including a brief statement on the use of decontamination methods. For instance, for studies with larger sample size, it may be possible to use Decontam (Davis et al., 2018) (frequency method) over the metagenomic OTU table to see if any taxa exhibit the property of being a contaminant (i.e., an inverse relationship with total DNA amount). This way, taxa suspicious of being a contaminant could be excluded from further analyses (e.g., pop genomics).

R1.4 response. We agree with the reviewer that one should be extra vigilant of potential contamination in ancient metagenomic datasets, and that decontamination techniques might provide an additional layer of security on top of traditional methods of ancient DNA validation. We included a statement on L.203-205 in the discussion, but also acknowledge that these methods are not appropriate for our dataset given the low sample size, but should be considered in future, larger-scale studies.

R1.5 • Third-generation sequencing: In line with what I wrote above, I suppose long reads could improve the quality of the data that could be obtained, despite how damaged the DNA is (e.g., long organelle contigs). Accordingly, I would also briefly discuss this point on what else needs to be explored (e.g., among 200-206 lines).

R1.5 response. Unfortunately, a switch in sequencing method to those capable of generating longer reads will not be beneficial for ancient DNA studies. Ancient DNA, by its nature, is both damaged and fragmented into smaller sequences³, often <100 base pairs (bp), which are generally short enough to be fully read by second-generation short read methods. In the case of our study, the majority of the *Nannochloropsis* aDNA is between 40-60 bp in length, with almost no reads longer than ~120 bp (Supplementary Figure S2-4, S7-8). Ancient DNA studies also usually require the sequencing of tens to hundreds of millions of reads, which is currently beyond the capabilities of third-generation technologies. Thus, third-generation sequencing methods are unlikely to be beneficial for ancient genomic assemblies.

Minor comments:

R1.6 • Line 118: Supplementary Figures S3 and S4.

R1.6 response. The Supplementary Figures reference in the main text have been changed to the correct numbers.

R1.7 • Lines 469-471: I would recommend making this easier to understand.

R1.7 response. We agree that the described method was difficult to follow. L.473-480 have been rephrased to clarify our terminology and the procedure. In addition, we added Supplementary Figure S13 to further illustrate how the method works.

R1.8 • The version of some bioinformatic software tools is missing in the main text, e.g., OGDRAW

R1.8 response. The version numbers for the OGDRAW (v1.77) and GeSeq (v1.3.1) tools were missing and have been updated accordingly (L.380-383).

Reviewer #2 (Remarks to the Author):

Overview: The study utilized two sediment cores to assess the presence of an algae genera in a lake at a single time point using modern molecular methods. The take home message is that their findings verifies the use of molecular approaches on sediment cores to assess historical community dynamics. While the molecular approach seems fine, the low sampling power and how the issue of sediment mixing should be addressed. The writing could also be improved as the paper is very organism specific at the moment and will have limited relevance to a wider audience. The bioinformatics pipeline to discern haplotypes itself holds some merit and may be a better focus instead of trying to fit the study into a palaeological context.

We thank the reviewer for their feedback.

We acknowledge that the study uses few sediment samples, which limits our potential to track larger, population specific, changes over time. The sedimentation issue is discussed in depth in Alsos *et al.* 2020². The two shotgun metagenomic samples are both from within the

sediment unit U2. A modified figure has been added to show where in the sediment cores these samples were taken (modified from Alsos *et al.* 2020²; Supplementary Figure S12), as well as highlighting the consistent detection of *Nannochloropsis* within unit U2 based on metabarcode data from both Alsos *et al.* 2020² and the present study (Supplementary Text; Supplementary Figure S12).

We agree that the manuscript had a strong organism-specific focus and have now made improvements to broaden its appeal (e.g. L.34-43, L.65-69). The methods presented here are potentially relevant outside palaeoecology, as it could help with all samples of degraded and/or mixed DNA. However, we keep some palaeoecological focus, to support the fact that our method was developed on challenging ice-age samples (due to short fragments). To streamline the manuscript, we now keep a more general focus throughout the main text and discussion, while still highlighting the age and type of material, whereas we provide more details about the palaeoecological context in the supplementary information.

R2.1 • Line 64-67: This is overly specific, there are several studies that have used molecular tools on non-targeted DNA to derive genomic diversity. The introduction should include a broader research focus to include the methods used.

R2.1 response. We agree with the reviewer that a broader metagenomic background is beneficial to the paper. The introduction has now been expanded with a paragraph describing the shotgun metagenomic analysis that is applied for microbial analysis as well as some background on population genomic analysis from complex samples (L.34-43).

R2.2 • Methods: There should be some text on how the possibility of false positives were taken into account with regards to the study design. The main issue being that taking a single DNA sample from each of the two cores leave a lot of room for random error in any downstream analyses.

R2.2 response. Although we have no internal replication in our data set, the fact that we can replicate our finding between two independent and (roughly) coeval samples indicates that our results are unlikely to be random sampling errors. In addition, we are unable to detect the Andøya shotgun-derived *Nannochloropsis* barcode sequences in any of the metabarcoding negative controls (Supplementary Table S10) or in a modern sediment metabarcode data set from the same lake (Supplementary Table S12). Based on this information, along with the presence of ancient DNA damage, we rule out that our detections are due to false positives. We attempted to detect other eukaryotic taxa in our data set (see Figure 1), but these detections were determined to be false positives based on our use of 'exotic' genomes to provide baseline data for a false positive detection rate. These points are now included and elaborated in the discussion (L.190-205).

R2.3 • Results: A lot of methods text is present that distracts from the results. I understand the format is difficult, but the section needs to be revised to focus on results and not methodology. Presently, it is not clear what the results were. Additionally, several parts are interpretative and should be moved to the discussion. E.g. lines 93-104, 106-110, 111-120, etc....

R2.3 response. We agree with the reviewer that several sentences in the results were leaning too much towards either methods or discussion. We have now reduced method details in each paragraph's introductory sentence to a brief method signpost and moved all inference/discussion points to the discussion. The sentences on L.89-91, L.121-122, L.134, L.171-175 and L.186-188 (note: these are the old line numbers) have been removed as they either duplicate the methods or discussion text. The sentences on L.84-87 and L.106-108 have been moved to the discussion, now L.186-189 and L.183-186 respectively.

R2.4 • Discussion: I would suggest focusing on the haplotype diversity analyses. The abundance/community interpretations are not well supported given the sampling design.

R2.4 response. We agree that the abundance and community interpretation is not well supported by our data and detracts from the general data, haplotype, and quality discussion. We have addressed this by removing the abundance discussion points from our manuscript and moving the more limited palaeoecological interpretation (L.215-222) to the Supplementary Text L.76-87.

R2.5 • Line 207-222: There is no statistical variation about the sampling points to infer differences in abundances of the current data set. The high abundance is just as likely to be a random sampling error.

R2.5 response. We agree that our data does not allow us to properly determine abundance. This point has now been removed from the discussion.

R2.6 • Line 234-237: Again, there are several existing studies using similar methods.

R2.6 response. We are aware of several modern studies and techniques that recover population genomic information from complex biological samples, such as microbiomes. Examples are: Truong *et al.* 2017, Albanese and Donati 2017, Lloyd-Price *et al.* 2017, Quince *et al.* 2017, Nurk *et al.* 2017 and Francis *et al.* 2019⁴⁻⁹, which we have now included in the introduction and discussion. However, it is not straightforward to apply these methods to sedaDNA samples, given the damaged and fragmented nature of the DNA. These points are now raised both in the introduction (L.34-43) and the discussion (L.228-230).

Reviewer #3 (Remarks to the Author):

Dear authors,

The manuscript „Environmental reconstruction of an Ice Age algal population“ authored by Youri Lammers, Peter Heintzmann and Inger Alsos presents a study about the identification of chloroplast and mitochondrial genomes of the algae *Nannochloropsis* detected in shotgun (metagenomic) data from two late glacial sediment samples from a lake in Northern Norway. The manuscript is well-written and technically convincing. The application of metagenomics on ancient lake sedimentary DNA is still a relatively novel approach, with only a few studies published so far. This study adds new

knowledge about late glacial taxonomic composition in a northern lake, which seems largely restricted to a few, but dominant taxa during this time period.

General comments:

The study gives support for a dominance of algae growth during the late glacial because the majority of classified reads are assigned to *Nannochloropsis*. The authors applied a robust taxonomic assignment which supports the authenticity of detected ancient DNA reads. The authors also point out that two different samples originating from a similar time slice, but from different sediment cores are highly similar in terms of taxonomic assignment. So far, the manuscript focuses on taxonomic classification and the identification of haplotypes from metagenomic data and these analyses seem very convincing. Beyond this, I am wondering if the authors could add additional information to the main manuscript, which is:

We thank the reviewer for their comments.

R3.1 • provide an overview (e.g. pie chart for sample A and B) for unidentified and classified reads (incl. tax. assignment) - this would give a nice overview and one does not need the Supplementary tables to understand the data basis

R3.1 response. We agree with the reviewer that this information could be graphically displayed. We expanded Supplementary Figure S1, which contains the metagenomic assignments, to include a pie chart that conveys the proportions of unidentified and classified sequences from two million reads for each sample.

R3.2 • indicate which parts of the chloroplast or mitochondrial genome of *Nannochloropsis* are possibly overrepresented in the ancient reads or are not covered at all (did you see an effect of PCR duplicates, which might have happened during library amplification in the indexing PCR?)

R3.2 response. Each organellar genome was fully reconstructed with all bases covered (stated on L.125-129), although some coverage fluctuation was observed over the full assembly, as is expected from any shotgun datasets (illustrated in Figure 2 and Supplementary Figure S5). During mapping and filtering of the data, any sequences that were identified as PCR duplicates were removed, as stated in the Methods section (L.357 and L.408), and not included in coverage calculations.

R3.3 • is it possible to say if mitochondrial or chloroplast DNA is better preserved? What is known about the abundance of mitochondria and chloroplasts in *Nannochloropsis*?

R3.3 response. Both organellar genomes have comparable coverage, with 64-62x for our chloroplast and mitochondria respectively, as stated in the results, which suggests comparable relative proportions of chloroplast and mitochondrial DNA in the data. Unfortunately, no organellar abundance data is available for modern *Nannochloropsis* species, meaning that we are not able to further investigate the conservation question. We note that the ratio of chloroplasts to mitochondria in living *Nannochloropsis* from high

latitudes is likely to fluctuate depending on season. In terms of other measures of DNA preservation, such as DNA fragment size and read termini deamination frequency, these are near-identical between chloroplast and mitochondrial DNA in both samples (Supplementary Figures S7 and S8).

R3.4 • Does your data support also evidence for nuclear *Nannochloropsis* DNA?

R3.4 response. We were able to detect nuclear *Nannochloropsis* DNA in our sediment samples. The results presented in Figure 1 and Supplementary Table 3 are based on data mapped against nuclear genomes. Furthermore, we were able to reconstruct the nuclear 18S and ITS markers, for which we presented the results in Table 2 and Supplementary Figure S9.

R3.5 • Are barcode regions (like ITS, 18S, *rbcl*) represented in the ancient reads? Could one use the metagenomic data and compare with metabarcoding?

R3.5 response. We were able to reconstruct the ITS and 18S nuclear barcodes and the *rbcl* plastid marker from our shotgun metagenomic data. Phylogenetic placement of the reconstructed markers is presented in Supplementary Figure S9 and summarized in Table 2. These full-length barcodes are not commonly used for metabarcoding of environmental DNA (eDNA), given their longer lengths and the fragmentary nature of eDNA. The metabarcode data that we presented in our study is based on the shorter plastid p6 loop marker, located within the *trnL* UAA intron¹⁰. The p6-loop barcode is present in our reconstructed chloroplasts, however, this marker lacks the resolution to identify all *Nannochloropsis* to species level, and thus is not useful for more elaborate taxonomic identifications for this genus (Supplementary Table S16).

R3.6 • Sample A seems to perform better than B, because it shows an overall higher read count and longer mean read length, does this affect the taxonomic assignment of the data? What might be the reason for this difference?

R3.6 response. We indeed observe a difference in both the quantity and length of the read data between the two samples. However, we note that the difference in average read length of the filtered data is only 3 bp. Further, both samples behaved similarly for our various analysis, including the metagenomic identification, the mapping based identification, phylogenetic placement of the organellar genomes, and barcode regions, suggesting that these differences have not affected our results. The exception is the haplotype estimation, which indicates a slightly higher proportion of three haplotype variants for sample Andøya_LGM_A, compared to Andøya_LGM_B. This is most likely due to the difference in sequence lengths, where a higher length allows for more data to be utilized for haplotype estimations. This point is now also included in our haplotype discussion (L.212-214).

There are multiple potential causes for the read length differences between our samples, such as relative DNA preservation or technical fluctuations during DNA extraction and/or

library preparation. The difference in read counts likely arose during library pooling prior to sequencing.

R3.7 • It is interesting that the authors detected *Mycobacterium avium* in their data. Could there be any connection between the adjacent bird cliff and the occurrence of *Mycobacterium avium* in the sediments? It is also known that *Mycobacterium* can produce resting cells (see Wu et al. Developmental transcriptome of resting cell formation in *Mycobacterium smegmatis*) and survive in soils for many days (Tio-Coma et al. "Detection of *Mycobacterium leprae* DNA in soil: multiple needles in the haystack"). Maybe these features enhance DNA preservation over time?

R3.7 response. Although we agree that the presence of *Mycobacterium avium* would be consistent with the adjacent bird cliff, the data unfortunately does not allow us to determine which *Mycobacterium* taxon is present in our data. All four *Mycobacterium* strains that were included in our genomic reference panel had roughly comparable mapping rates of 1.1-1.7 thousand sequences per megabase of reference genome (kseq/Mb) before removal of conserved sequences and 0.6-0.9 kseq/Mb when only considering unique sequences (Supplementary Table S3). Based on these mapping results we cannot confidently identify which *Mycobacterium* taxon (or taxa) are present in our datasets.

We see no evidence for improved DNA preservation for *Mycobacterium* in our ancient DNA dataset. Both the mapped sequence fragment length distributions and the cytosine deamination rates are near-identical between *Nannochloropsis limnetica* and *M. avium* (Supplementary Figures S3 and S4).

R3.8 • Although the authors show typical ancient patterns in the *Nannochloropsis* and *Mycobacterium* reads, I completely miss the investigation of extraction blanks and library controls. Please explain how you used controls during all preparation steps (from DNA extraction to final sequencing reads) to monitor possible contamination sources. If you also sequenced negative controls, I would highly recommend to present the results in the manuscript.

R3.8 response. Extraction blanks were taken, but were only subjected to metabarcoding of the *trnL* p6 loop barcode region. We have now included this metabarcode data set in the Supplementary Text and Supplementary Figure S12. The two shotgun-derived *Nannochloropsis* barcodes were not detected in the extraction and PCR blanks (Supplementary Table S10). However, we did detect a third barcode in five samples and four controls, but these were scattered throughout the cores, not detected in the shotgun data, and not replicated between the PCR replicates. We don't consider our *Nannochloropsis* detections to be a result of contamination, due to the presence of ancient DNA damage patterns and the lack of detection in the controls for this experiment or previous metabarcoding datasets originating in the same lab (see also responses R1.1 and R2.2). Although the metabarcode primers are not able to amplify *Mycobacterium*, the fact that both damage patterns and fragment length distributions are near-identical to our *Nannochloropsis* detections suggests that the DNA molecules have the same preservation history, as expected from them being deposited in lake sediment during the last ice age. We

have further justified that our data are not the result of contamination on L.190-205 and Supplementary Text L.58-74.

Detailed comments:

R3.9 • The authors might think about to change the very general title of the manuscript in a more specific one, because the promised “environmental reconstruction” is only a very little part in the manuscript. Maybe you use a title like: “Nannochloropsis organelle genomes retrieved from a metagenomic analyses of glacial lake sediments in Northern Norway”.

R3.9 response. The current title does not refer to environmental reconstruction, but instead to palaeogenomic reconstruction from environmental DNA data, as opposed to conventional reconstructions from body fossils. Whilst we can see how this might be confusing to someone expecting an environmental reconstruction, we clearly use the phrase ‘environmental palaeogenomics’ in the abstract. We prefer the brevity of the current title for broad appeal and note that most of the keywords in the reviewer’s suggested alternative are present in the abstract.

R3.10 • Line 242

Please indicate shortly if the core was stored frozen or at 4°C.

R3.10 response. We updated our method section to clarify that our cores are stored at 4°C prior to sampling (L.247-248).

R3.11 • Line 262

How did you concentrate the sample? Did you use a kit? Please specify.

Line265

How many PCR cycles were used in the indexing PCRs?

R3.11 response. We have updated the methods section with both details on the DNA concentration procedure and the number of indexing PCR cycles (L.265-269).

R3.12 • Line269

2x80 cycles, you might lose DNA fragments which are very long, and can’t merged after sequencing? Do you think this might affect your data? Please specify the average fragment length of your libraries

R3.12 response. The maximum merged read length for our 2 x 80 bp paired-end data is 145 bp (after allowing for a 15 bp overlap). We have now expanded Supplementary Table S1 to distinguish between reads that were discarded and those that were unmerged (see also R3.13). The percentages of the data that were unmerged are 11% and 26%. However, we detect almost no DNA fragments that approach 145 bp, based on alignment to human, *Nannochloropsis*, and *Mycobacterium* genomes (Supplementary Figures S2-4, S7-8), suggesting that the number of DNA fragments alignable to these genomes and >145 bp, and therefore excluded, is negligible for both samples. This is expected as ancient DNA is highly fragmented into short fragments³, often <100 bp.

R3.13 • Line 275

Please specify the number of unmerged reads with good quality. Give the relative abundance of merged and unmerged sequences based on the raw read count.

R3.13 response. Our merged, discarded, and remaining unmerged read numbers in Supplementary Table S1 have been expanded to give a more detailed breakdown of why a read pair was discarded. Either a read could not be merged (because it was longer than 145 bp or from a repetitive region), or the resulting merged read did not pass our minimum length criteria of 35 bp, which includes the removal of adapter dimers.

R3.14 • Line 703

Please provide the unit of the y-axis in Figure 1c. I guess for algae and bacteria it is total reads and for human and others % of filtered reads?

R3.14 response. Our plotted numbers in Figure 1c are the number of mapped sequences per megabase of reference, as indicated by the axis title in the Figure. We have moved the current title so that it is less likely to be confused for a general figure title and explained the data further in the figure caption.

R3.15 • Line 221

Supplementary Figure 11 is cited, but not present in the Supplement. Should be 9, I guess.

R3.15 response. Good catch. The cited Supplementary Figure S11 in the main text was correct, however it was incorrectly listed as Supplementary Figure 9 in our supplement. All figure and table numbers have been updated and checked after the inclusion of additional supplementary information.

R3.16 • Supplementary Figure S9: Overview of *Nannochloropsis* detections, using 10 sedaDNA...

Please indicate if these results were retrieved by metabarcoding or metagenomic data, and if metabarcoding was used, please indicate which metabarcode has been applied. For example, the metabarcoding approach using the P6 loop (Taberlet et al. 2007) is not designed for the detection of algae in sediments, which will likely produce false negatives. Typically, the amplicon for *Nannochloropsis* using the P6 loop metabarcode is very short (about 12bp) and shows low identities to the reference in the databases. For this reason, I am not sure, if the detection obtained from non-specific metabarcoding will help to conclude on a distribution pattern through time. I am also not sure, what is the conclusion out of this compilation for the shot gun data?

R3.16 response. We agree that Supplementary Figure S11 can be improved by including the method used for the identifications. The sequencing and identification methods (either shotgun metagenomics (S) or metabarcoding (M)), has been added to the figure for each study presented.

Although the *trnL* p6-loop marker is not designed to amplify algae, the number of mismatches between the primers and *Nannochloropsis* is small enough, with 3 and 2

mismatches for the forward and reverse primer respectively, that amplification is possible. The *Nannochloropsis* p6-loop barcodes are relatively short, but are unique within reference databases and not overlapping with any other known algae (even those closely related to *Nannochloropsis*) or vascular plants. We consider the identifications trustworthy, especially for the two barcodes that match the p6-loop variants derived from our shotgun metagenomic data. However, we note that one of the *Nannochloropsis* barcodes (identified as *N. granulata*) does seem to share a haplotype with an unknown contaminant. These points are now all discussed in the Supplementary Text (L.58-74).

The conclusion of the meta-analysis is that, regardless of the methods used, *Nannochloropsis* is not unexpected from ancient sedimentary DNA and seems to have a cosmopolitan geographic and temporal distribution (L.179-181 and Supplementary Text L.77-87). The *Nannochloropsis* results obtained for our Andøya data are thus in line with previous studies.

References

1. Alsos, I. G. et al. Plant DNA metabarcoding of lake sediments: How does it represent the contemporary vegetation. *PLoS One* **13**, e0195403 (2018).
2. Alsos, I. G. et al. Last Glacial Maximum environmental conditions at Andøya, northern Norway; evidence for a northern ice-edge ecological 'hotspot'. *Quat. Sci. Rev.* **239**, 106364 (2020).
3. Thomsen, P. F. & Willerslev, E. Environmental DNA – An emerging tool in conservation for monitoring past and present biodiversity. *Biol. Conserv.* **183**, 4–18 (2015).
4. Truong, D. T., Tett, A., Pasolli, E., Huttenhower, C. & Segata, N. Microbial strain-level population structure and genetic diversity from metagenomes. *Genome Res.* **27**, 626–638 (2017).
5. Albanese, D. & Donati, C. Strain profiling and epidemiology of bacterial species from metagenomic sequencing. *Nat. Commun.* **8**, 2260 (2017).
6. Lloyd-Price, J. et al. Strains, functions and dynamics in the expanded Human Microbiome Project. *Nature* **550**, 61–66 (2017).
7. Quince, C. et al. DESMAN: a new tool for de novo extraction of strains from metagenomes. *Genome Biol.* **18**, 181 (2017).
8. Nurk, S., Meleshko, D., Korobeynikov, A. & Pevzner, P. A. metaSPAdes: a new versatile metagenomic assembler. *Genome Res.* **27**, 824–834 (2017).

9. Francis, T. B., Krüger, K., Fuchs, B. M., Teeling, H. & Amann, R. I. Candidatus Prosiliicoccus vernus, a spring phytoplankton bloom associated member of the Flavobacteriaceae. *Syst. Appl. Microbiol.* **42**, 41–53 (2019).
10. Taberlet, P. *et al.* Power and limitations of the chloroplast trn L (UAA) intron for plant DNA barcoding. *Nucleic Acids Res.* **35**, e14–e14 (2007).

Modified and new figures

Figure 1: (a, b) Location of Lake Øvre Æråsvatnet (circled in red, panel B) in the ice-free refugium of Andøya in northwest Norway. The regional ice extent for Scandinavia in panel A has been plotted for 22 (outer) and 17 (inner) kcal yr BP and is based on Hughes *et al.*⁶⁸. The local ice extent in panel B is plotted for 22-18 and 18-17.5 kcal yr BP and is based on Vorren *et al.*³¹. (c) Taxonomic composition of the LGM Andøya sediment samples, based on alignment to a reference panel of 42 eukaryotic or bacterial nuclear genomes. **The scale is the number of sequences mapped to each megabase (Mb) of the reference genome.** For readability, the algal, bacterial, and human results are plotted separately with differing y-axis scales. **Change: Slightly moved the y-axis legend in (c) to avoid confusion with a general figure title. Added additional caption text (coloured red) to further explain the numbers plotted. Change made as a response to reviewer 3 (R3.14).**

Supplementary Figure S1: Visualization of results from the BLAST-based metagenomic analysis. (a) The proportion of assigned reads for each sample.

(b) The metagenomic breakdown of the taxa present. All tips are collapsed to the genus-level or higher, with the exception of the two most read-abundant tips: *Mycobacterium* and *Nannochloropsis*. The numbers represent the reads assigned to each node for samples Andøya_LGM_B and Andøya_LGM_A respectively.

Change: The inclusion of the pie charts at the request of reviewer 3 (R3.1).

Supplementary Figure S11: Overview of *Nannochloropsis* detections, using 10 *sed*aDNA data sets either as previously published or based on reanalysis of available data. Plotted on the Y-axis is the inferred abundance of *Nannochloropsis*; 3: dominant in the period, 2: common, 1: rare and 0 is absent. Data were binned into 5000 year time periods, which might obscure finer patterns. No *sed*aDNA data was available for time periods marked with an asterisk. Studies used either shotgun metagenomics (S) or metabarcoding (M) for *Nannochloropsis* detection. **Change:** For each study it is indicated if the detections are based on shotgun metagenomics (S) or metabarcoding (M) at the request of reviewer 3 (R3.16).

Supplementary Figure S12: Detection of *Nannochloropsis* in the Lake Øvre Æråsvatnet, Andøya, Norway cores based on the metabarcode data presented in this study and that of Alsos *et al.* 2020. The metabarcode data, lithological units, and loss-on-ignition curves are visualized along the core images. For each *Nannochloropsis* p6-loop barcode, the proportion of PCR replicates that contain the variant is displayed. All detections are coloured blue or orange for this study or Alsos *et al.* 2020 respectively, while non-detections are black. The two shotgun metagenomic samples; Andøya_LGM_A and Andøya_LGM_B are marked. Figure is adapted from Alsos *et al.* 2020. **Change: A new figure to illustrate the overlap between our shotgun and metabarcoding detections for *Nannochloropsis*. This figure was added to provide additional evidence for our *Nannochloropsis* detections, see reviewer comments R1.1, R1.3, R2 (general comment), R2.2 and R3.8.**

Supplementary Figure S13: Visualization of the haplotype calling method. (a) Variant window selection. (b) Retrieval of candidate reads. (c) Scoring of the haplotypes present in the window. **Change:** A new figure to help explain our haplotype calling method. Included as part of our response to reviewer 1 (R1.7).

Reviewers' comments:

Reviewer #1 (Remarks to the Author):

The authors did a great job of addressing my comments, and I have no further comments to make. Moreover, I especially like the (now) stronger focus on the novel haplotype estimation method.

Reviewer #2 (Remarks to the Author):

Overall, the study is a combination of classical sediment core analyses with modern genomics methods to infer interspecific species diversity. The introduction could still benefit from more detail pertaining to the background surrounding the genomic population analyses. The results are limited with regards to sampling replication (spatial sampling $N=2$), so should be interpreted cautiously, though the temporal dataset ($N = 153$; unknown structure) might be investigated much more rigorously (I am now quite confused why these data were not looked at using some sort of time series analyses and why the information is kept in the supplement). Parts of the figures/tables in the supplement could have been incorporated with the main text to present a more convincing argument, but it is rather unclear with the current legends and text what the intended flow and direction of the study is (clear objectives in the introduction would help here).

This may be simply a different or focused interest between myself and the authors, for which I apologize. There seems to be good potential with the data, but I am struggling to see the execution.

Introduction - The utilized methods are not developed for this study; therefore, they are not novel as such. What has been done is an application of existing methodology, to understand natural population structure. There is no indication of study objectives or hypotheses for the study for the reader to rely upon for further reading.

Methods - The genomics protocol is fine, if a bit verbose. The lack of spatial sampling makes any population-based analyses difficult to interpret, though the indication of temporal sampling in the supplement might be something worth investigating further. Additional information regarding how the data were assessed after the bioinformatics is needed to support the main findings.

Results: The number of samples and subsamples from each core needs to be clearly stated. Part of this information is in the supplement, which should be in the main text.

Raw read counts are not necessarily indicative of population abundances. This can be argued even more so for low samples sized sediment data which may simply be an effect of localized events. There are also inherently different numbers of chloroplast /mitochondria across the different species.

From Supplementary Table S4, and other supplementary legends, it is unclear how the data were combined (e.g. summation or mean across extracts)

I do not see Tables S1, S5, S8, S12 or S13 in the review files

Discussion - There is a drive to use environmental based DNA (e.g. sedaDNA, eDNA, what have you), which is fully appreciate. While the specific application of extracting DNA, using specific inserts, and using specific bioinformatic approaches among studies may imply unique applications, they are not in themselves novel. The general approach taken for the study is interesting, however, I would greatly appreciate, as a reader, a more open discussion regarding the interpretation of the population level

analyses, which are not presented here, in respect to other environmental DNA based and classical population genomics studies instead of the current focus emphasizing the highly specific novel aspects of the current study.

Reviewer #3 (Remarks to the Author):

The revised manuscript „Environmental palaeogenomic reconstruction of an Ice Age algal population“ authored by Youri Lammers, Peter Heintzmann and Inger Alsos presents a study about the identification metagenomes of the algae *Nannochloropsis* detected in shotgun (metagenomic) data from two late glacial sediment samples from a lake in Northern Norway. The authors revised the manuscript according to the reviewer’s suggestions, which improved the structure and focus of the manuscript. The reviewers required more evidence to proof the authenticity of the *Nannochloropsis* DNA data in old sediment samples. The authors explained that they used ancient DNA patterns to proof the authenticity of the data and run controls along with metabarcoding approach. The metabarcoding approach however is different from shot gun methodology and both approaches require controls to indicate putative contaminations during the lab work when working with ancient DNA. In contrast to comparable studies on paleogenomic analyses (e.g. Slon et al. 2017), the authors did not run extraction or library controls along with their shot gun sample preparation, which weakens the overall study design.

Reviewer 3 made the following remarks:

The revised manuscript “Environmental palaeogenomic reconstruction of an Ice Age algal population“ authored by Youri Lammers, Peter Heintzmann and Inger Alsos presents a study about the identification metagenomes of the algae *Nannochloropsis* detected in shotgun (metagenomic) data from two late glacial sediment samples from a lake in Northern Norway. The authors revised the manuscript according to the reviewer’s suggestions, which improved the structure and focus of the manuscript. The reviewers required more evidence to proof the authenticity of the *Nannochloropsis* DNA data in old sediment samples. The authors explained that they used ancient DNA patterns to proof the authenticity of the data and run controls along with metabarcoding approach. **The metabarcoding approach however is different from shot gun methodology and both approaches require controls to indicate putative contaminations during the lab work when working with ancient DNA. In contrast to comparable studies on paleogenomic analyses (e.g. Slon et al. 2017), the authors did not run extraction or library controls along with their shot gun sample preparation, which weakens the overall study design.**

We thank reviewer 3 for pointing out that this was still not clear. We have now modified the manuscript to demonstrate beyond any reasonable doubt that our results are not caused by contamination.

The **widely-used gold standard for ancient DNA authentication** is assessing cytosine deamination profiles, and we note that reviewer three presents no objection to this. Our shotgun metagenomic data mapped to *Nannochloropsis* and *Mycobacterium* exhibit clear cytosine deamination patterns. Crucially, these are almost identical between an algal and bacterial taxon, indicating a similar preservation history and source. This precludes a sole modern origin for the DNA assigned to these taxa. This is in contrast to the sequence data identified as human, which lack such damage patterns. To highlight these critical authentication data, we now include the cytosine deamination profiles for *Nannochloropsis*, *Mycobacterium*, and human in a new manuscript figure, with the full damage plots still available in the supplementary information.

We are aware that Slon *et al.* 2017¹ sequenced their laboratory negative controls. The focus of Slon *et al.* was to retrieve trace amounts of ancient hominin DNA using shotgun metagenomics and hybridization capture. The risk of modern human contamination is severe in such an experiment, given the ubiquitous presence of modern human DNA in both ancient DNA laboratories and reagents². Therefore, even though damage patterns demonstrate that there is ancient DNA present, they do not preclude a mixture of ancient DNA and modern contamination. The same risk is true for microbial metagenomics of, for example, microbiomes from ancient dental calculus³. In such cases with high risk of modern contamination, laboratory negative controls are useful to track and correct for background

trace contamination. However, the focus taxon of our study, *Nannochloropsis*, is not a known laboratory contaminant. We further note that laboratory negative controls are not helpful if the sample itself is contaminated.

Let us consider two alternative possibilities: that our data (1) are partially contaminated by modern or young ancient DNA or (2) were the result of contamination in the laboratory by an unknown source of ancient DNA that included *Nannochloropsis* and *Mycobacterium*. These could have occurred during coring, core sampling, DNA extraction, or library preparation. First, *Nannochloropsis* is absent from the study lake today⁴ and it is only sporadically detected from 14,500 cal BP onwards (Supplementary Figure S12; Alsos *et al.* 2020⁵). This makes scenario one highly unlikely and additional laboratory negative controls would not provide further insight. Nevertheless, we have now considered the possibility that one of our organellar genomic variants may represent modern contamination that was otherwise masked by the global assessment of cytosine deamination profiles. We therefore re-ran our variant consensus pipeline on only reads exhibiting cytosine deamination. We were able to recover 64-70% of the previously identified variant sites for both organellar genomes, demonstrating that they are of ancient origin, with the remaining sites dropping out due to lowered coverage (Supplementary Table S8; Lines 166-167, 199-201, 476-484).

Shotgun metagenomics and metabarcoding are complementary, independent methods to assess the presence of a taxon's DNA within a DNA extract. Although metabarcoding typically targets longer DNA fragments that may not be preserved (e.g. Ziesemer *et al.* 2015⁶, Murchie *et al.* 2020⁷), this is not the case for *Nannochloropsis*, which has a short *trnL* p6-loop amplicon length of 51 base pairs (including primer binding sites) that is close to the mean DNA fragment length of our shotgun libraries. Critical to a comparison between the two methods is that both are able to distinguish our two chloroplast genome high and low frequency variants. Therefore, even though our DNA extraction negative controls were only processed using metabarcoding, we can still use them to validate whether or not *Nannochloropsis* is present as a contaminant in these controls. Despite replication, the two *Nannochloropsis* chloroplast genome variants were not detected in any DNA extraction (and PCR) negative controls. Nor were these sequences detected in the negative controls by Alsos *et al.* 2020⁵. We have now inserted the relevant summary data table into Figure S12 to clearly show this. Further, metabarcoding success of the two variants shows clear correlation with sediment lithology (detected throughout the U2 and U3b lithological units; Figure S12), which is not expected were the DNA extracts contaminated by an exogenous source. We therefore find it extremely unlikely that there was a laboratory contamination event during DNA extraction.

This leaves the possibility of contamination during shotgun library preparation. However, if this was the case, then the libraries would have been contaminated by *Nannochloropsis* variants that correspond precisely to those found independently using metabarcoding of the same DNA extracts. We consider the chance of this coincidence to be highly unlikely and that a common-origin hypothesis (ie. *Nannochloropsis* ancient DNA originated from the sediment) is far more parsimonious.

In summary, we present multiple lines of evidence to suggest that our data are not the result of contamination; the presence of cytosine deamination profiles, a lack of *Nannochloropsis* in our DNA extraction controls, and the detection of the same sequence variants across two

independent approaches: shotgun metagenomics and metabarcoding. We have modified our manuscript to make this clear and changes include: Lines 166-167, 199-201, 476-484 (changes coloured in blue), a new Figure 2 and updated Figure S12 and Table S8). Thus, we address the concerns of reviewer 3.

We look forward to your reply and thank you for further consideration of our manuscript.

Yours,

Youri Lammers
Peter D. Heintzman
Inger G. Alsos

Updated figures and tables

Revised manuscript
Figure 2
Figure S12
Table S8

Literature cited

1. Slon, V. *et al.* Neandertal and Denisovan DNA from Pleistocene sediments. *Science* **356**, 605–608 (2017).
2. Llamas, B. *et al.* From the field to the laboratory: Controlling DNA contamination in human ancient DNA research in the high-throughput sequencing era. *STAR: Sci. Technol. Archaeol. Res.* **3**, 1–14 (2017).
3. Warinner, C. *et al.* A Robust Framework for Microbial Archaeology. *Annu. Rev. Genomics Hum. Genet.* **18**, 321–356 (2017).
4. Alsos, I. G. *et al.* Plant DNA metabarcoding of lake sediments: How does it represent the contemporary vegetation. *PLoS One* **13**, e0195403 (2018).
5. Alsos, I. G. *et al.* Last Glacial Maximum environmental conditions at Andøya, northern Norway; evidence for a northern ice-edge ecological ‘hotspot’. *Quat. Sci. Rev.* **239**, 106364 (2020).
6. Ziesemer, K. A. *et al.* Intrinsic challenges in ancient microbiome reconstruction using 16S rRNA gene amplification. *Sci. Rep.* **6**, 27163 (2015).

7. Murchie, T. J. *et al.* Optimizing extraction and targeted capture of ancient environmental DNA for reconstructing past environments using the PalaeoChip Arctic-1.0 bait-set. *Quat. Res.* 1–24.

REVIEWERS' COMMENTS:

Reviewer #3 (Remarks to the Author):

The revised manuscript „Environmental palaeogenomic reconstruction of an Ice Age algal population“ authored by Yuri Lammers, Peter Heintzmann and Inger Alsos present a study on ancient *Nannochloropsis* genomes detected by metagenomic approaches. The reviewers concern about the authenticity of the data because of missing laboratories controls. I appreciate, that the authors agree that this would be beneficial and would be of great support to prove the authenticity of data as indicated by this additional section in the revised manuscript (line 209-212). I also think that the mapdamage patterns of reads mapped to the *Nannochloropsis* reference genome show convincing ancient DNA patterns, which support the authenticity of the generated ancient DNA data from LGM lake sediments of Northern Norway, which silenced my concerns.